# Cellular interfaces with hydrogen-bonded organic semiconductor hierarchical nanocrystals

Mykhailo Sytnyk[1,2], Marie Jakešová[3,4,5], Monika Litviňuková[4], Oleksandr Mashkov[1,2], Dominik Kriegner[6], Julian Stangl[7], Jana Nebesářová[8], Frank W. Fecher[9], Wolfgang Schöfberger [10], Niyazi Serdar Sariciftci[3], Rainer Schindl[4,11], Wolfgang Heiss[1,2] & Eric Daniel Głowacki [3,5]

Successful formation of electronic interfaces between living cells and semiconductors hinges on being able to obtain an extremely close and high surface-area contact, which preserves both cell viability and semiconductor performance. To accomplish this, we introduce organic semiconductor assemblies consisting of a hierarchical arrangement of nanocrystals. These are synthesised via a colloidal chemical route that transforms the nontoxic commercial pigment quinacridone into various biomimetic three-dimensional arrangements of nanocrystals. Through a tuning of parameters such as precursor concentration, ligands and additives, we obtain complex size and shape control at room temperature. We elaborate hedgehog-shaped crystals comprising nanoscale needles or daggers that form intimate interfaces with the cell membrane, minimising the cleft with single cells without apparent detriment to viability. Excitation of such interfaces with light leads to effective cellular photostimulation. We find reversible light-induced conductance changes in ion-selective or temperature-gated channels.

[1] Materials for Electronics and Energy Technology (i-MEET), Friedrich-Alexander-Universität Erlangen-Nürnberg, Martensstraße 7, 91058 Erlangen, Germany. [2] Energie Campus Nürnberg (EnCN), Fürtherstraße 250, 90429 Nürnberg, Germany. [3] Linz Institute for Organic Solar Cells (LIOS), Physical Chemistry, Johannes Kepler University, Altenbergerstraße 69, 4040 Linz, Austria. [4] Institute for Biophysics, Johannes Kepler University, Gruberstraße 40, 4020 Linz, Austria. [5] Laboratory of Organic Electronics, ITN Campus Norrköping, Linköpings Universitet, Bredgatan 33, 60221 Norrköping, Sweden. [6] Department of Condensed Matter Physics, Charles University, Ke Karlovu 5, Prague 121162, Czech Republic. [7] Institute of Semiconductor and Solid State Physics, University Linz, Altenbergerstraße 69, Linz 4040, Austria. [8] Biology Centre of the Czech Academy of Sciences—Institute of Parasitology, Branišovská 31, České Budějovice 37005, Czech Republic. [9] Bayerisches Zentrum für Angewandte Energieforschung (ZAE Bayern), Immerwahrstr. 2, 91058 Erlangen, Germany. [10] Institute of Organic Chemistry, Johannes Kepler University, Altenbergerstraße 69, 4040 Linz, Austria. [11] Institute for Biophysics, Medical University of Graz, Harrachgasse 21/IV, 8010, Graz Austria. Mykhailo Sytnyk and Marie Jakešová contributed equally to this work Correspondence and requests for materials should be addressed to R.S. (email: rainer.schindl@medunigraz.at) or to W.H. (email: wolfgang.heiss@fau.de) or to E.D.G. (email: eric.glowacki@liu.se)

Biology at the single-cell level is true nanomachinery[1]. As we master new techniques for manipulating matter on the nanoscale, opportunities for interfacing with biological systems arise. Recently, nanomaterial interfaces at the cellular level have been demonstrated to achieve cell morphology control[2–4], cell fate determination[5, 6], sensing[7, 8], nanoinjection[9–11] and delivery of genetic material for transfection[12]. In all of these applications, interrogation of intracellular events relies on sharp high-aspect ratio nanostructures[9, 13, 14]. Artificial high-aspect nanostructures have similarly been a focus of interest for electronic interfacing with living cells, being sought after for applications in high-quality extracellular and intracellular electrophysiology[7, 15] recording and stimulation, and for providing a bridge into the cytosol for both delivery and intracellular sensing[10, 11]. Inorganic materials, especially silicon, and metals like platinum and gold predominate in all these applications. A common goal is getting as close an interface to the cell as possible, forming a minimal cleft, and ideally with large area[13, 15]. Optimising such structures is especially critical in the case of (opto)electronic interfaces, where the cleft between the cell and electronic element results in electric field screening and poor coupling[16–18]. Recently, close cellular interfaces with nanoscale amorphous silicon particles have been able to give reversible photostimulation of excitable cells[19]. Control of biology with light at the single-cell level is a concept with far-reaching consequences in both fundamental biological research and applied medicine. Optogenetics is widely considered to be one of the most significant development in neuroscience in the past decade, since it enables highly localised targeting at the single-cell level both in vitro and in vivo[20]. Its reliance on genetic transfection introduces challenges and limitations, however, which has motivated extensive exploration of nongenetic means of optical control. Several reports have shown the possibility to achieve light-induced manipulation of cells, particularly excitable cells, either mediated by light-absorbing particles[19, 21, 22], or thin-films[23–25], or using direct near-infrared optical heating[26]. In the past years, a growing spectrum of novel bioelectronics applications have been enabled by organic semiconductors, which have superior biocompatibility and mechanical properties, and novel functionality relative to silicon[27–29]. These features, combined with their high optical absorbance coefficient, have made nanoscale thin films of organic semiconductors suitable for

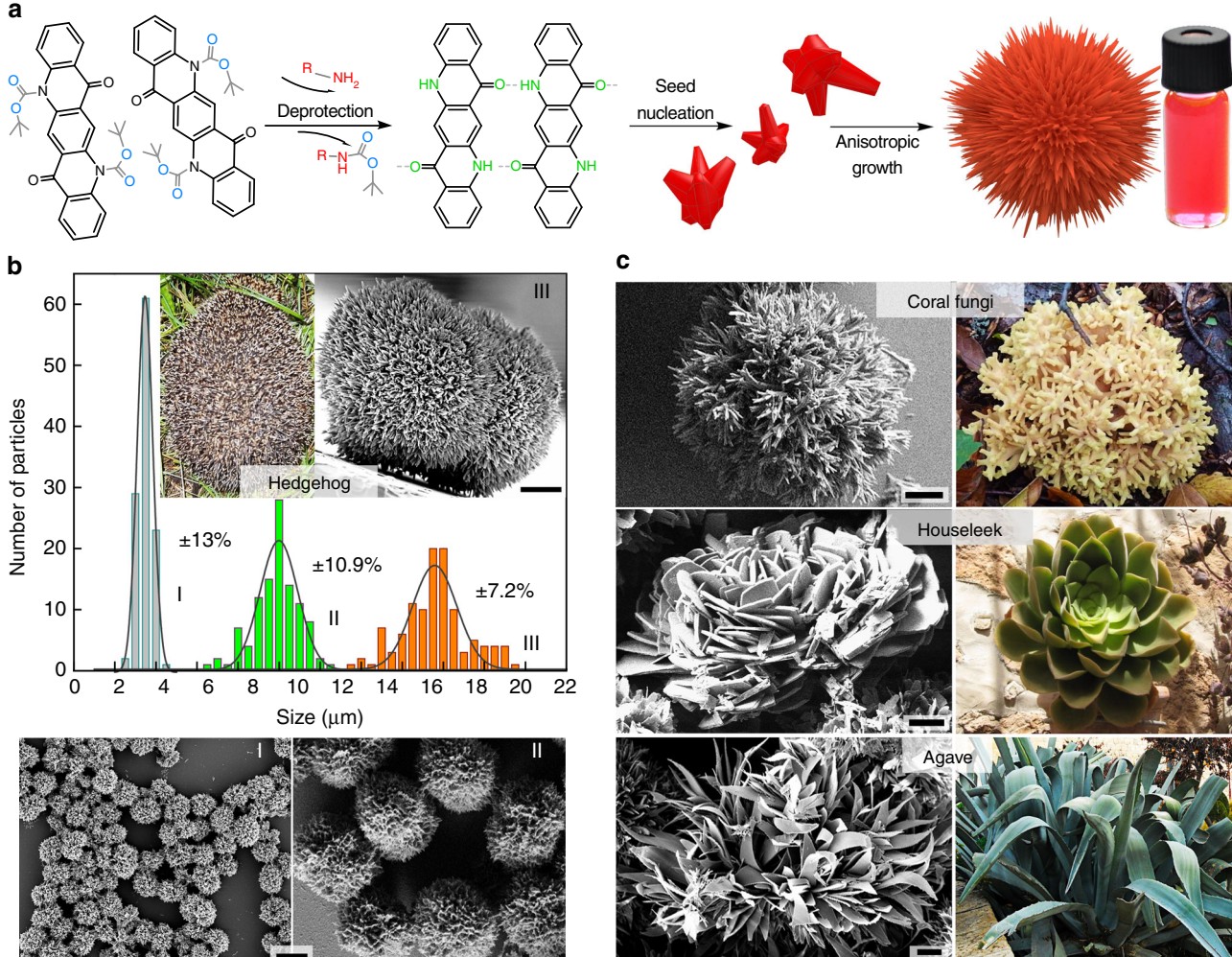

**Fig. 1** Colloidal quinacridone hierarchical nanoarchitectures with bio-inspired shapes. **a** The synthesis proceeds when soluble *N*,*N*′-di(*t*-butoxycarbonyl) quinacridone (tBOC-QNC) is deprotected via an amine migration reaction, giving monomeric quinacridone (QNC) molecules that crystallise via an interplay of H-bonding and π–π stacking. These ligand-covered colloidal nanocrystals then assemble due to van der Waals forces into hierarchical colloidal assemblies. **b** The size of hedgehog crystals is determined by the starting concentration of tBOC-QNC, allowing narrow size distribution to be achieved. Three sizes—I, II and III are shown in the SEM micrographs. **c** Nanoarchitectures with shapes of various plants, given on the photos, are obtained by using different ligands. Scale bars on the electron micrographs = 4 μm

optoelectronic photostimulation of single cells[30–32] and retinal tissues[25, 33]. The issue of the cell/semiconductor cleft still remains an obstacle for these organic devices, however.

The starting point of our work is the desire to create a new family of organic semiconductor structures that can by virtue of morphology form an intimate contact with the cell membrane. To this end, we develop a synthetic method to yield hierarchical colloidal architectures comprising organic semiconductor nanocrystals. We synthesise these using a ligand-mediated approach, not only to afford fine synthetic control of the structure, but also to yield a crystal surface modified with a ligand monolayer suitable for favourable interaction with lipid bilayer cell membranes. As an organic semiconductor suitable for biointerfacing, we choose quinacridone (QNC), a nontoxic magenta-coloured pigment industrially produced primarily for inks and paints[34]. We present methods whereby QNC hierarchical assemblies form upon ligand-mediated QNC-precursor decomposition at room temperature followed by nucleation and assembly into hierarchical structures. By manipulation of conditions such as initial precursor concentration, reaction time, solvent, and chemical additives, we control size, shape, and crystalline polymorphism of the QNC structures, yielding spherical shapes consisting of high aspect-ratio nanocrystals with forms reminiscent of hedgehogs. These hedgehog colloidal semiconductors, with overall diameter similar to a eukaryotic cell (10 μm), can be used directly in cell culture. We find that two cultured cell lines used routinely in electrophysiology experiments, rat basophilic leukaemia (RBL), and human embryonic kidney (HEK) cells, grow preferentially on such hierarchical nanocrystal structures, forming close interfaces with minimal cleft after a few hours in culture. This occurs without apparent changes in cell viability. The hierarchical assembly has the auspicious property of being able to mechanically deform under the growing cell, resulting in an interface with a minimal cleft. This occurs because, though the constituent crystals are rigid, the hierarchical superstructure is held together by van der Waals forces and is thus plastic. These single cell/semiconductor nanostructure interfaces lend themselves to patch clamp electrophysiology experiments. Visible-light photoexcitation of the semiconductor nanostructures leads to reversible changes in ion conduction through ion-selective channels (K$^+$ rectifiers) and temperature-gated ion channels in cells growing on the hedgehogs. We see both rapid changes on the order of a few milliseconds as well as longer-scale ion conductivity increases, created by an interplay of rapid photoinduced changes in membrane capacitance and photothermal heating. Our work demonstrates a promising new platform for optoelectronic interfaces with living matter.

## Results

**Hierarchical nanoarchitecture syntheses.** Nanocrystalline architectures of organics can be grown using various vapour or solution deposition/evaporation techniques[35–37]. Colloidal techniques[38, 39] are prevalent in the case of hierarchical inorganic materials[40–44]. We have recently introduced the idea of ligand-mediated syntheses of colloidal organic monocrystals[45] using a range of hydrogen-bonded pigments. Here we make hierarchical crystals for cellular interfaces by designing a new colloidal synthetic method. As a molecular building block, we choose QNC. The hydrogen-bonded pigment QNC is particularly interesting in the context of biological applications due to reported nontoxicity[46] and presence of NH functional groups that enable direct bioconjugation reactions[47]. Recently QNC, in the form of vacuum-evaporated thin films, has been reported as a promising semiconductor, with outstanding stability, ambipolar charge carrier mobility, and favourable optoelectronic[48] and

photocatalytic[49] properties. The multifunctionality and availability of QNC make it a good target for making organic semiconducting hierarchical nanostructures. Our chemical route (Fig. 1a) for QNC colloidal nanoarchitectures relies on first transforming the insoluble QNC pigment powder, obtained from a paint supplier, into a soluble dye, N,N′-di(t-butoxycarbonyl) quinacridone (tBOC-QNC), using the known amine protection reaction[50] with t-butoxycarbonyl (tBOC). QNC, due to interplay of intermolecular hydrogen bonding between carbonyl and amine functional groups and π–π stacking, is insoluble. By interrupting the hydrogen bonding with tBOC functionalisation of the NH group, a highly soluble dye is obtained[45, 50]. The tBOC group can be removed by heat or strong acids[51]. To accomplish our crystal growth at room temperature and under mild conditions, we discovered a new deprotection reaction: the ability of carbamate esters to migrate between amine groups. The reaction can occur at room temperature, as the tBOC carbamate ester unit will favour migration to the amine that is the stronger nucleophile. Depending on the reactivity of the amine added to the tBOC-QNC, this reaction can go to completion in a few minutes (for highly reactive amines like methylamine) or even from hours to weeks (see Methods and Supplementary Table 1). We found that primary amines with long aliphatic chains interact at moderate rates with tBOC-QNC, resulting in reactions lasting several hours. Depending on the type of amine used, the starting concentration of tBOC-QNC, and the presence of solvents and chemical additives that selectively interact with the carbonyl or amine functionality on QNC, a range of hierarchical nanostructured microcrystals can be grown (Fig. 1b, c). When tBOC-QNC is reacted with oleylamine (acting as both solvent and reactive amine) at room temperature, 3D hierarchical structures with the shape of hedgehogs, consisting of self-assembled nanoneedles with diameters smaller than 50 nm (Fig. 1b) are reproducibly obtained. While such complex structures are relatively hard to synthesise in the case of inorganic materials, here we obtained them with narrow size dispersion in the range of ±10% by simply varying the starting concentration of tBOC-QNC. Changing the ligand species results in additional bio-inspired shapes of the final nano-architectures: while oleylamine provides hedgehogs, methylamine gives a coral fungi architecture consisting of dendritic nanocrystals (Fig. 1c). Mixtures of butylamine and di-methylaminopyridine give 2D nano-platelets arranging into nanoflower architectures, with a shape reminiscent of houseleek plants. Pure butylamine as ligand results in agave-shaped nanoarchitectures of approximately similar dimensions. In all cases, the microcrystals demonstrated outstanding colloidal stability when transferred to organic solvents (e.g. chloroform or chlorobenzene) due to extensive capping with alkylamine ligands. The structural integrity was found to be very robust in solution for several months, even after treatments by ultrasonication. Using X-ray diffraction, we can prove that all the QNC nanoarchitectures in Fig. 1c consist of differently sized and shaped nano-units having the same internal crystal structure (γ) (Supplementary Fig. 1), even though QNC pigments are known to crystallise in four polymorphs (α$_1$, α$_2$, β, γ)[52].

**The growth mechanism.** In order to understand the mechanisms behind the anisotropic growth of the colloidal crystals, first the aspects of chemical reactivity were probed. A beautiful aspect of this reaction is that it can be easily monitored in situ by optical absorbance spectroscopy, thus the concentration of the constituents can be extracted via the Beer–Lambert law. Optical absorption measurements reveal three different stages linked to a chemical state of the QNC molecules (Supplementary Fig. 2): at the beginning of the reaction (stage 1) the QNC is

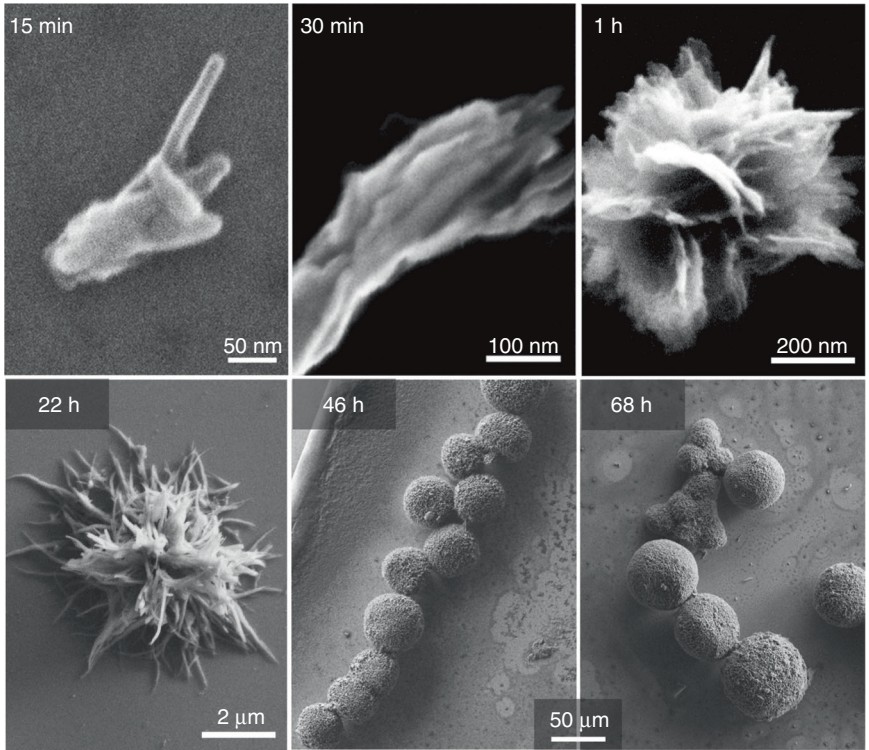

**Fig. 2** SEM imaging of hierarchical crystal growth. Aliquots removed at different times during synthesis reveal the growth process of hierarchical crystals. Branching is occurring already at 15 min, and continues as the microstructures grow over 68 h. Monomeric quinacridone molecules precipitate onto exposed facets of the crystals, generating growth along this facet that progressively ceases to grow due to increased steric hindrance caused by attached alkylamine ligands. Repetition of this process maintains branched growth as the microstructures expand spherically

double-protected by tBOC, in stage 2 it is mono-tBOC protected and in stage 3 it is completely unprotected and in the monomeric state constituting the building block of crystals. The double-protected QNC shows an absorbance peak at 420 nm, the monoprotected at 470 nm and the unprotected at 520 nm. After stage 2 there is almost no signal from tBOC-QNC and the dominant species in solution is monomeric (peak at 510 nm). Based on these time-dependent optical measurements (Supplementary Note 1 and Supplementary Figs. 2–4), we conclude that the amine-induced deprotection of tBOC-QNC involves chemical migration of a tBOC group, via a pseudo-first-order chemical reaction in conditions where the concentration of the reactive amine is significantly greater than that of tBOC-QNC. The migration of the tBOC group to the reactive amine is confirmed by identifying $t$-butoxycarbonyl alkyl amide products with nuclear magnetic resonance spectroscopy (Supplementary Figs. 5 and 6). This amine-induced migration is significantly different from the well-established tBOC deprotection reactions, namely acid-catalysed and thermal, because in these two cases irreversible decomposition of the tBOC groups occurs. Those reactions are relatively harsh, requiring either strong acids like trifluoroacetic acid or temperatures in excess of 120 °C. A clear advantage of amine-induced deprotection is its occurrence at room temperature. The second advantage is a flexible control of the reaction kinetics by the tBOC-QNC concentration and reactivity of the amine and thus crystallisation and growth control of nanocrystals. To understand the growth over time, we removed aliquots from the reaction with oleylamine, which proceeds at a moderate rate, and imaged them using scanning electron microscope (SEM) (Fig. 2). It is clear that even at the earliest stages of the synthesis needle-like nanocrystallites branch off of nucleation points. Over time the branching continues, resulting in comb-like aggregates of nanocrystals that grow into spherical microstructures. The

spherical shape indicates that nucleation and growth occurs in colloidal solution rather than on a surface, as molecular monomers must nucleate from all sides of the microstructure to account for the spherical forms (Supplementary Figs. 7 and 8).

**Polymorphism control**. So far, we have discussed control of hierarchical structure shape by using different ligands while conserving the internal molecular crystal structure ($\gamma$). We further found that by manipulating reaction time and chemical additives, phase-pure samples of three polymorphs could be prepared: $\alpha_2$, $\beta$ and $\gamma$ (Fig. 3a, b). The three crystallisation routes are illustrated in Supplementary Fig. 9. The speed of crystal growth and coordination of QNC monomers determines the final crystal structure. Amine-induced deprotection lasting several hours reliably produces $\gamma$ as described previously. The $\gamma$ phase features a 'criss-cross' lattice of QNC molecules where each molecule forms single NH···O= hydrogen bonds to four neighbours. Slowing down the reaction to last over several days (1–10 days) by using a noncoordinating solvent and a lower amine concentration gives the $\alpha_2$ polymorph, qualitatively similar to $\gamma$ but with closer spacing between chains of hydrogen-bonded QNC molecules and lower triclinic symmetry (Supplementary Fig. 10). Finally, by using a solvent that coordinates with amine groups, for example cyclohexanone, a completely different hydrogen-bonding pattern between QNC molecules is obtained, namely a linear-chain arrangement where each QNC molecule hydrogen bonds to only two neighbours, forming H-bonded sheets in the $\beta$ polymorph (Supplementary Fig. 11). A major benefit of polymorphism control is that the different crystal structures lead to distinctive optical properties (Fig. 3c, d). We find that each polymorph is luminescent, with excitonic emission that is remarkably narrow compared to previous findings of solid state thin-film luminescence in QNC, which report coexistence or even dominance of

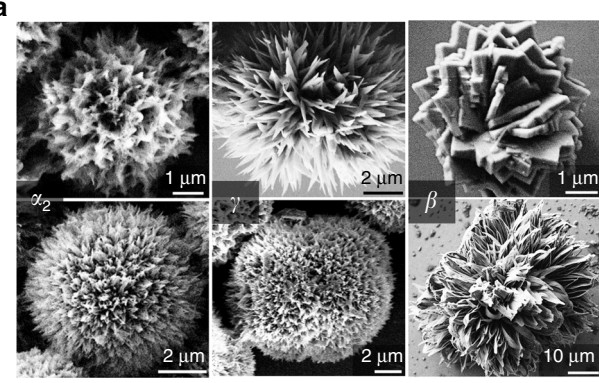

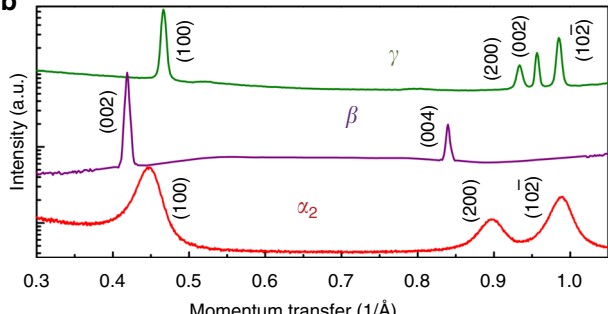

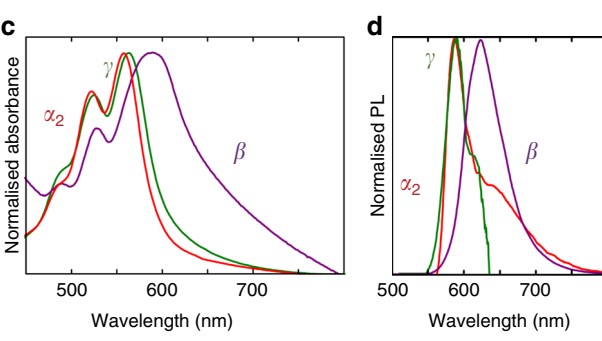

**Fig. 3** Control of internal crystal structure. Polymorphism can be controlled by modulating reaction time and coordinating chemical additives. **a** SEM micrographs of $\alpha_2$, $\beta$ and $\gamma$ crystals. **b** X-ray diffraction, confirming the phase-pure quality of the colloidal samples, **c** absorbance and **d** luminescence of each sample. The upper row of SEM images shows the hierarchical nano-dagger modification, while the lower row shows hierarchical nanoneedle assemblies

broad defect state luminescence in the region of 700–850 nm[48, 53]. This signifies that the crystalline semiconductor quality of the hierarchical crystals is high, and apparently far less defects are present than in vacuum-evaporated films. Two hierarchical hedgehog structural schemes could be observed for each polymorph: Shorter alkylamines like butylamine favour formation of nanodagger hedgehog crystals consisting of triangular units of tens to hundreds of nanometres in width tapering down to sharp tips (upper row of images in Fig. 3a). Using bulkier amines such as oleylamine result in hedgehogs consisting of nanoneedles (lower row of images in Fig. 3a).

**Interfaces with single cells**. From the outset of this study, we surmised that the organic needle-like hierarchical QNC hedgehogs should be capable of forming close and high surface-area contacts with cells. To evaluate this, we first drop-cast colloidal solutions of QNC hedgehogs on glass substrates suitable for cell culture. The two types of $\gamma$ hedgehogs shown in Fig. 3a were used:

hierarchically assembled arrays of nanoneedles, while the second is the nanodagger modification, featuring nano-scale triangular daggers tapering down to a sharp 10–20 nm wide point. Drop-cast hedgehogs were found to adhere well to both hydrophobic octyltriethoxysilane (OTS)-modified, as well as hydrophilic (3-aminopropyl)triethoxysilane (APTES)-modified glass substrates, and did not delaminate in cell culture solutions, or after UV sterilisation. Rat basophillic leukemia (RBL) and human embryonic kidney (HEK) cells were chosen for culture due to their utility in electrophysiology experiments. Cell culture was carried out for up to 3 days, with samples removed for SEM imaging and cell viability assays. Glass substrates modified to be favourable for the given cell culture (OTS-modified glass for RBL cells and APTES-modified for HEK) were used for SEM imaging. The RBL cells were found to grow on the QNC planar films prepared by vacuum sublimation; however, HEK cells showed no attachment. On the other hand, we found that RBL and HEK cells readily attach to both types of hedgehogs. RBL cells form remarkably conformable interfaces with the nanostructured surface of hedgehog crystals already after 1.5 h in culture, apparent from SEM (Fig. 4). The cell/crystal interfaces can be conveniently viewed also by optical microscopy, where the crystals' luminescence allows fluorescence imaging of the semiconductor structures (Supplementary Fig. 12). From SEM imaging, it is apparent that the plasma membrane and extracellular matrix conforms to the nanoneedle structures, causing anchoring of cells onto the microhedgehog (Fig. 4). The evaluation of the true interface cleft distance in vivo is a question of current debate, as the fixation of cells for electron microscopy can produce a cleft morphology different from what is present in vivo[15, 54]. The cleft distance has been determined to change as much as 10–50 nm following fixation in some cases, even where cryofixation methods are employed. In the case of SEM imaging of hedgehog/cell interfaces, the inevitable 'artefact problem' of cell shrinkage occurring during fixation actually yields interesting information: we observed frequent examples where the hedgehogs are actually pulled apart by cells, with cells extracting nanoneedles from the parent microstructure, or sometimes splitting the hedgehog structure between multiple cells (Fig. 4c, Supplementary Fig. 13). This indicates that the membrane/nanoneedle interaction can be stronger than the forces holding the hierarchical architecture together. Over longer times in culture, the cells are found to transition from a rounded shape to spread more extensively on the hedgehogs (Fig. 4d). Based on viability assays (CytoTox-Glo luminescence-based assay, Supplementary Fig. 14) we conclude that the cells remain viable while having such an extensive contact area with the crystals. Carried out over 3 days, the assay demonstrated no difference in viability between cells cultured with hedgehogs, planar or powder QNC and control samples. This experiment suggests that neither form of the QNC material is acutely cytotoxic. QNC itself, as a commercial pigment, has been studied with regards to consumer safety and determined to be nontoxic[46]. Like RBLs, HEKs form a high interfacial area contact with hedgehog crystals, (Fig. 5), with occasional examples of engulfment-type processes[15] clearly occurring (Fig. 5b). As mentioned before, HEK cells do not attach to planar evaporated QNC films. To discriminate the role of hierarchical nanostructure vs. QNC surfaces themselves, we prepared samples with hedgehogs drop-cast onto a glass slide, followed by sublimation of a uniform 80 nm thin film of QNC over the entire sample area, including on top of the hedgehogs. HEK cells were found to grow exclusively on the hedgehogs and not anywhere on the planar films (Fig. 5c, Supplementary Fig. 15). From this it is clear that hedgehog structures promote cell attachment and growth due to their nano-microstructure[3]. HEK cells progressively spread over hedgehog structures during

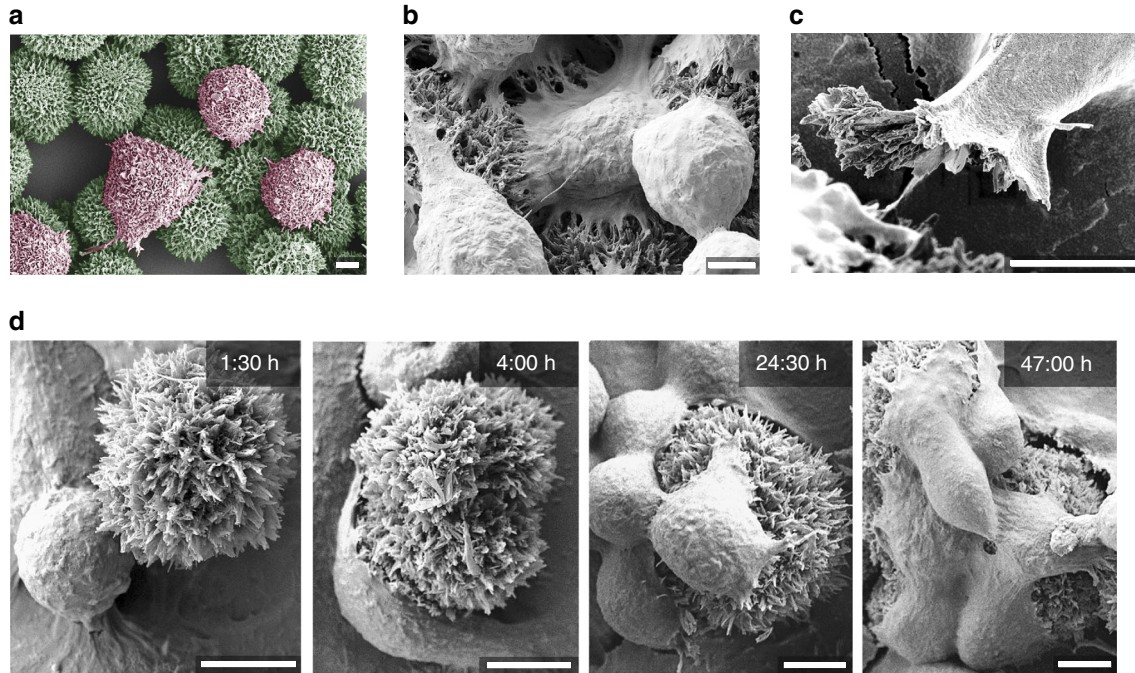

**Fig. 4** SEM imaging of rat basophilic (RBL) cells cultured with quinacridone nanohedgehogs. **a** RBL cells cultured on 10-μm-size nanoneedle hedgehogs, with cells false-coloured *pink* and hedgehogs false-coloured *green*. The image captures the cells when they are degranulating. **b** Tight interfaces between RBL cells and nanoneedle hedgehogs. **c** The cell membrane and extracellular matrix of RBL cells can rip nanoneedles out of the hierarchical microcrystal, demonstrating the strength of the cell/nanocrystals interaction. **d** RBL cell culture interrupted at different intervals (hours shown in image *upper right corners*), showing the progression of cell attachment, proliferation and morphology over time. Scale bars = 5 μm

cell culture (Fig. 5d), with uncompromised viability verified by assays (Supplementary Fig. 14). Cross-sectional images revealing the cell/semiconductor interface were possible with HEK cells since, serendipitously, strain occurring during the fixation/dehydration procedure could lead to splitting of the cell/hedgehog pair, conveniently revealing the morphology of the interface (Fig. 6). It must be noted that since the fixation procedure leads to shrinkage in these samples, this imaging can only give us the lower limit for the actual cleft size. Keeping this in mind, two critical observations can be made: first, there appears to be no evidence that the nanocrystals penetrate through the cell membrane, rather, the cell membrane arranges flush with the nanocrystals, with extremely close contact and no distinguishable cleft. Second, the hierarchical nanocrystalline structure partially collapses, buckles and distorts to give way under the cell. Even though the extent of the distortions revealed by SEM is a result of fixation stresses, we find no examples of nanocrystallites breaking the integrity of the cell membrane. This leads us to conclude a unique mechanical advantage—the hierarchical structure has the mechanical freedom to move, and the constituent nanocrystallites can be rearranged by the growing cell. This interpretation is consistent with the observation discussed previously of RBL cells ripping nanocrystallites out of the hierarchical parent structures. This behaviour can easily be rationalised considering that the QNC molecules in the nanocrystals are held together by a strong interplay of $\pi$–$\pi$ stacking and hydrogen-bonding, while the hierarchical arrangements are held together by much weaker van der Waals forces. These findings offer a new paradigm for organic semiconductors for bioelectronics—hierarchical architectures can show mechanical plasticity while being constituted of a rigid and stable material.

**Photostimulation of ion channels**. Having established stable and close, high-surface area cell/semiconductor interfaces, we used patch clamp electrophysiological techniques to probe the effect of

visible light irradiation (532 nm) on the cells. Our first electrophysiology measurements were on the endogenous potassium inward rectifier channels expressed in RBL cells (Fig. 7a). The $K^+$ inward rectifier channels are of general interest since they are responsible for maintaining the resting membrane potential of many types of cells in animals, bacteria and plants. In whole-cell voltage–clamp measurements, we delivered through-objective illumination to patched cells (532 nm, 10 ms pulses, with three energies 10, 30 and 50 μJ). The illuminated area had a 15 μm diameter. The $K^+$ inward rectifier was selectively measured using a linear voltage-ramp protocol (Fig. 7a). Patched cells were recorded first in the dark and then with pulsed laser illumination (10 ms pulses). In the dark, reproducible rectifier current-voltage characteristics are obtained, indicating no non-specific membrane leakage. The integrity of patched cells was found to be identical for controls growing on plain glass (which show no photoinduced effects) in comparison with those on hedgehog structures. For cells growing on hedgehogs, application of light pulses generates depolarising currents at the beginning of the light pulse, with a corresponding spike in the opposite polarity at the end of the light pulse. This behaviour is visible at all voltage values along the sweep. At voltage values more negative than −80 mV, where the $K^+$ inward rectifier is open, following the transient current spike there is an increased inward current plateau (Fig. 7a, inset). After illumination, $K^+$ current returns to the baseline, dark, value. This net reversible increase in inward $K^+$ current scales with the light energy dose (Fig. 7b, inset) and is higher for nanoneedle-type hedgehogs compared with nanodagger ($n = 11$–13 cells). Control cells, growing on the same substrates but not in direct contact with hedgehogs, show no photoinduced changes whatsoever ($n = 20$ cells). Continuous wave (CW) illumination over longer time-scales comprising several whole voltage sweeps (hundreds of milliseconds with lower intensities of light, 5 mW/mm²) demonstrated that the $K^+$ rectifier curve does not change its current–voltage characteristics, only the net inward current peak

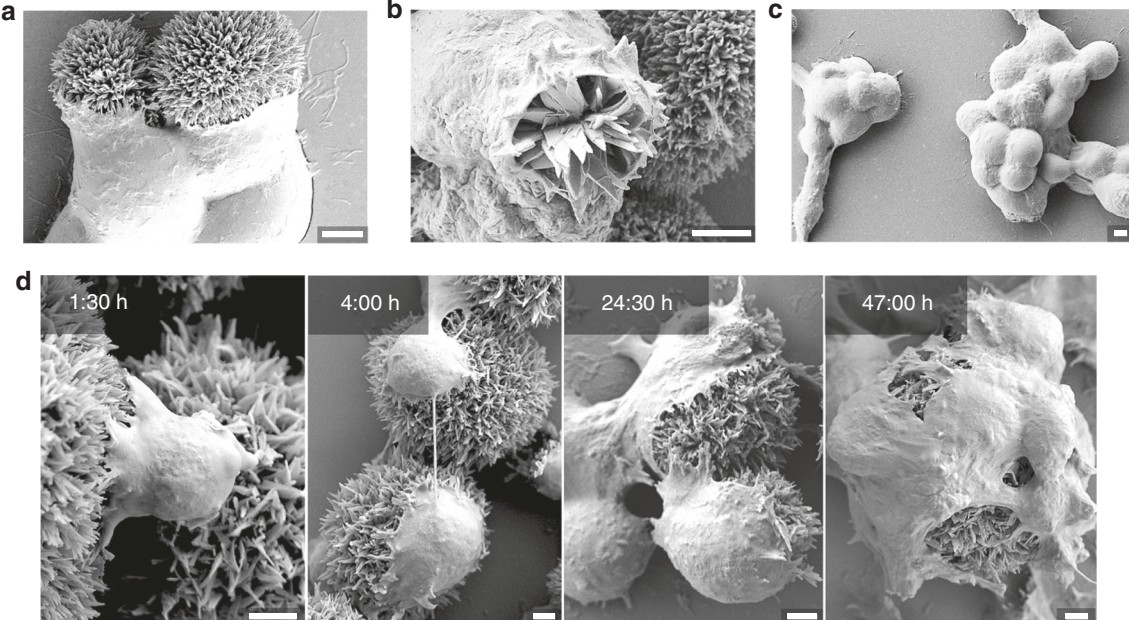

**Fig. 5** SEM imaging of human embryonic kidney (*HEK*) cells cultured with quinacridone nanohedgehogs. **a** Close and highly conformal interfaces form after a few hours in culture. **b** Engulfment-like events are observable in the case of HEK cells, here a cell is imaged in the course of an endocytosis event with a nanodagger hedgehog. **c** HEK cells do not grow on evaporated quinacridone thin-films. Here cells are cultured on substrates with hedgehogs where the entire surface, including hedgehogs, is coated with a uniform sublimated quinacridone thin film, thus demonstrating that the nanostructure and not chemical nature of the substrate is critical for cell adhesion. **d** Interface contact between HEK cells and hedgehogs increases over cell culture time, and morphology transitions from rounded cells with a few attachment points to cells showing extensive spreading on the hedgehog structures. Scale bars = 2.5 μm

reversibly increases with illumination (Fig. 7b). We now consider the two separate light-induced observations: First, the rapid depolarising current spikes present throughout the trace, and second, the increased current 'plateau' region that is apparent for time-scales from 10 to 1000 ms. Our observation of depolarising current can be explained by a rapid photothermal or photocapacitive effect (Fig. 7e). Photocapacititve effects could arise from charging of the QNC surface with negative charges and capacitive coupling[55], resulting in transient depolarisation of the cell membrane. Recently the phenomenon of fast heating-induced capacitive currents induced by amorphous silicon microparticles in contact with the plasma membrane has been reported by Jiang et al.[19]. There, the photothermal capacitance change gives rise to transient depolarisation of the membrane. These findings were in line with earlier work that demonstrated that intense infrared pulsed illumination could lead to capacitive membrane depolarisation[26]. Martino et al.[31] in the case of polymeric thin films observed apparent photothermal depolarisation behaviour, but preceded by a more rapid photocapacitive depolarisation[31]. In principle, our observation of depolarising current spikes can be explained by a rapid photothermal or photocapacitive effect. Whatever the origin, the capacitive peaks cause only a passive membrane response and no clear effect on K+ rectifier conductance. Considering the 'plateau' increase region: it is observable only in the voltage range where the K+ channel is open, at both short and longer time scales, coupled with no observation in changes in current–voltage characteristics, suggesting that photothermal heating leads to increased ion diffusion rates and therefore higher current through the channels when they are open. This effect is reversible in both short (10 ms) and longer (hundreds of ms) time-scale regimes. With the evidence generally pointing towards the presence of photothermal heating, we elected to transfect HEK cells with a temperature-activated channel: the transient receptor potential vanilloid (TRPV1) ion channel, famous for its role in producing the 'hot' taste of chilli peppers by being sensitive to capsaicin, and also transduction of pain caused by heat. Voltage-ramp measurements for TRPV1-transfected HEK cells grown on hedgehogs resulted in a more outward rectifying current–voltage relationship and gave qualitatively identical photoinduced behaviour as the K+ channels, namely transient depolarising current spikes and increased plateau regions at voltages where the channel normally shows activity (Fig. 7c). At more positive voltage polarisations, the light-induced capacitive depolarisation becomes smaller, qualitatively similar to results found for rapid infrared photothermal cell excitation[26]. One critical difference is apparent, however: at the cell resting potential (−60 mV) a reversible photoinduced cation influx current occurs (Fig. 7d). This demonstrates that temperature-gated channels can be directly and rapidly photostimulated in cells under normal physiological conditions. These results complement nicely recent findings of photoinduced stimulation of Ca2+ current in TRPV1 channels using nanoparticles of semiconducting polymers[22]. While that study demonstrates changes on the time scale of hundreds of milliseconds, using the patch clamp technique with our hedgehog crystals we are able to observe more rapid and reversible changes in TRPV1-mediated current. Finally, the picture that emerges is one where a photothermal mechanism can increase the ion flow through open channels without otherwise changing current-voltage characteristics, though there is a concurrent presence of faster depolarising current behaviour (Fig. 7e). Photothermal heating is unambiguously behind the light-modulated electrophysiology behaviour we observe in the potassium inward rectifier and in the temperature-gated TRPV1 channels. Rapid and localised heat transfer to cells has been targeted by various applications both in vitro and in vivo[19, 26, 31], and clinical applications of optical neural stimulation for neuroprosthetics are currently actively explored[56]. The potential success of the hierarchical organic crystal architecture in this context may rely not only on the formation of the close and high surface-area

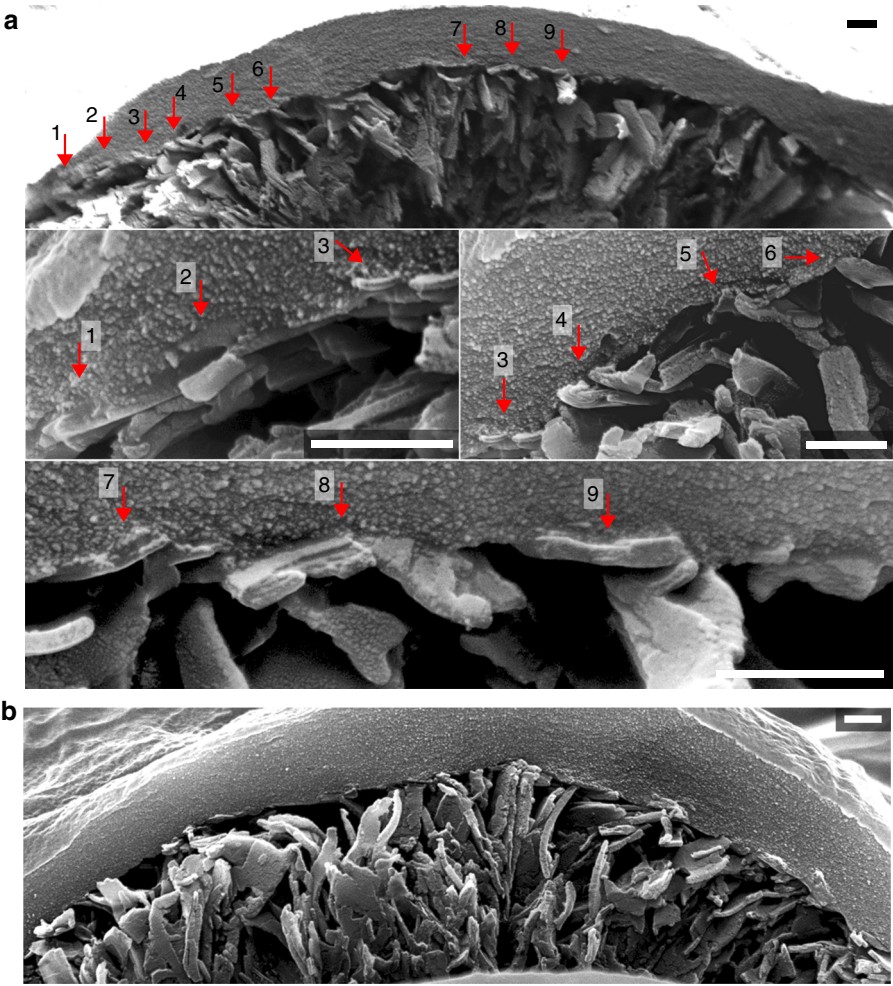

**Fig. 6** Cross-sectional interface of cells and hedgehogs imaged with SEM. **a** An interface between a HEK cell and nanodagger hedgehog, with close contacts between cell membrane and nanocrystallites labelled and shown in higher magnification. **b** Cross-section overview of another HEK/nanodagger interface, clearly showing the extent of distortion and crumpling of the nanodagger assembly under the cell. Scale bars = 500 nm

interface, but also on the specific photothermal behaviour inherent for such structures. We simulated the photothermal heating of hedgehogs vs. spheres and planar films, using the known absorption coefficient for QNC and assuming only convective heat dissipation to surrounding water. The high aspect ratio needles of hedgehogs heat up to higher temperatures faster than the surface of a planar film or a similar micron-scale sphere (Supplementary Fig. 16). The hierarchical semiconductor nanoarchitectures shown here, by virtue of being of similar size and shape to single cells, function as a platform for inducing photoeffects in a highly local manner, providing means of stimulating cells without invoking the need for genetic modification with light-sensitive ion channels.

## Discussion

In this work, we have introduced a room-temperature colloidal synthetic methodology for obtaining size-controlled and shape-controlled nanocrystalline hierarchical assemblies of the organic semiconducting building block QNC. These organic semiconductor hierarchical nanoarchitectures offer unique advantages for next-generation bioelectronics interfaces at the single-cell level. They feature the attractive properties of high aspect-ratio nanostructures that were to-date the domain of inorganic materials, notably silicon, but are mechanically much more pliable since the rigid nanocrystals are held together by van der

Waals forces. This enables very close interfaces with cells to form, with a minimal cleft. The minimisation of the cleft is a key parameter for several bioelectronics applications, especially electrophysiological recording and stimulation. QNC, as an organic semiconductor, has a high absorbance coefficient in the visible region, and we demonstrate that light irradiation of cells growing with a tight interface on QNC hierarchical nanocrystals results in photostimulation effects. Using pulsed irradiation, we are able to reversibly increase current flow through $K^+$ inward rectifier channels, at both millisecond and second time scales. A photothermal heating mechanism is implicated as most critical, as the hierarchical nanocrystals can efficiently and rapidly heat the interface with the cell. We exploit this feature by reversibly photostimulating the opening of temperature-gated channels (TRPV1) at normal physiological conditions. These demonstrations open up many potential research directions on this novel type of material. Future research should focus on establishing nanostructure-to-function relationships of how cellular function can be affected and controlled, and in particular should exploit the mechanical deformability of such assemblies. The details of such interfaces, as well as the occurrence of endocytosis-like processes, can be studied conveniently with optical means by leveraging the advantage of the strong luminescence of QNC combined with standard immunostaining procedures. Semiconductor-mediated optical stimulation of cells, despite encouraging seminal reports on the topic, remains largely

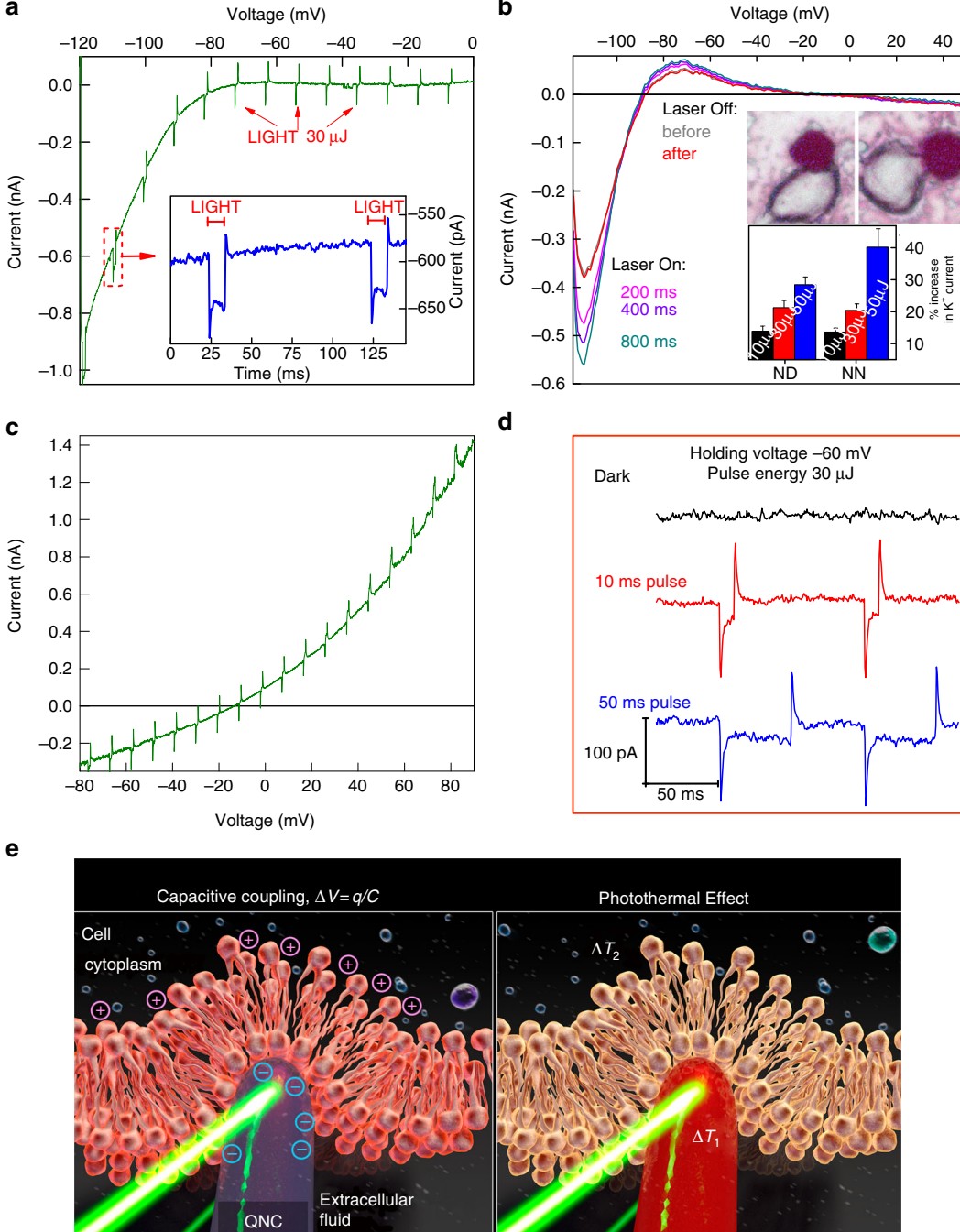

**Fig. 7** Hedgehog-mediated photostimulation of ion channels. **a** Reversible photostimulation of the K$^+$ inward rectifier channel with 10 ms light pulses in an RBL cell grown on a 10-μm-size nanoneedle hedgehog apparent in voltage–clamp measurements in whole-cell configuration. Rapid (<1 ms time scale) light-induced depolarising currents are visible over the entire voltage sweep range, however only where the K$^+$ inward rectifier channel is open ($V < -80$ mV) are there photoinduced increases (+30%) in K$^+$ inward current. The *inset* shows the effects of light pulses, at a holding potential of −110 mV with the depolarising current spikes and a plateau region of K$^+$ current increase, the latter attributed to local photothermal heating and faster ion diffusion. **b** The K$^+$ rectifier in an RBL cell on hedgehog measured using CW illumination over 1 s, 5 mW/mm$^2$. The current–voltage characteristic remains the same, only the peak value of K$^+$ current increases reversibly with illumination. This is consistent with photothermal heating raising the local temperature of the cell, increasing the current of ions through the open channels. The *inset* shows statistics for photoinduced K$^+$ influx increase percentage (K$^+$ rectifier) of the illuminated RBL cells growing on nanoneedle-type and nanodagger-type hedgehogs. Error bars represent the standard error of the mean ($n = 11$–14). The photomicrograph shows how cells in contact with hedgehogs appear under the microscope. **c** Current–voltage relationship of cation current in temperature-sensitive TRPV1 channels overexpressed in HEK cells on hedgehogs. Depolarising current spikes are visible throughout the whole trace while increases in cation current are apparent in voltage ranges where the channel is open. **d** HEK cell with TRPV1 give rapid and reversible photoinduced stimulation of cation influx when measured at the cell resting potential, −60 mV. **e** Schematic illustration of the mechanisms possible at nanocrystal/cell interfaces. Photocapacitive membrane depolarisation can occur on a short time-scale (*left*) and rapid photothermal changes can lead to capacitive modulation at short time scales, while increases in through-channel conductance due to localised photothermal heating (*right*) can be observed on time scales from 10 ms up to 1000 ms

unexplored. We believe that the use of simple organic pigments as light-absorbing particles will allow many researchers to enter into this field. For in vitro electrophysiology, the interfaces presented here are already suitable and highly advantageous, since we obtain clear single-cell level reversible stimulation that is localised. Our finding of photostimulation of the cell/semiconductor interface yielding reversible electrophysiological responses should be built upon to enable devices that give stimulation via true electronic mechanisms. This can be achieved by creating a donor/acceptor core-shell structure, which can generate charge separation with high photovoltage and surface-charge density—the two key benchmark parameters urgently sought to enable true photocapacitive cellular stimulation. Using organic crystal engineering such structures should be accessible. Finally, for application in retinal implants, a colloidal solution of nanocrystal assemblies can be sub- or epiretinally injected, and their size and shape similarity with rod and cone cells combined with favourable cell adhesion and interface-forming properties can make them a potent artificial retinal implant technology. We argue that pigments like QNC are an obvious materials choice for a biocompatible organic semiconductor: a cheap pigment that we use today for cosmetics and printing inks can provide an interface between the information age and us.

## Methods

**Materials and characterisation**. The commercially available QNC pigment was obtained from Kremer Pigmente or from TCI and was used as-received. All other chemicals were obtained from Sigma-Aldrich and used without additional purification. X-ray diffraction patterns were measured using synchrotron radiation at beamlines BM20/ESRF, Grenoble and powder diffraction beamline P02 at Hasylab Hamburg with 11.5 and 60 keV X-ray photons, respectively.

**QNC latent pigment synthesis**. tBOC-QNC was prepared by mixing QNC pigment powder (3.75 g, 12 mmol) in anhydrous tetrahydrofuran (600 ml) kept under nitrogen atmosphere at room temperature with di-*tert*-butyl dicarbonate (*t*-Boc$_2$O, 12.644 g, 58 mmol) and 4-dimethylaminopyridine (2.93 g, 24 mmol). This mixture was stirred for 48 h and monitored by thin layer chromatography. The crude solution was evaporated almost to dryness and filtrated in a chromatography column through a 80-fold amount of silica gel, with a 19/1 mixture of toluene/ethyl acetate (AcOEt) to obtain crystalline products in yields up 60%. This product was further purified by recrystallisation from AcOEt prior to nanocrystal synthesis. Sample purity was verified by comparison to literature spectra[45].

**Anisotropically grown nanoarchitectures**. QNC nanoarchitectures were obtained by room-temperature decomposition of tBOC-QNC in the presence of primary amines. To yield the smallest hedgehogs shown in Fig. 1b, 20 mg (39 μmol) of tBOC-QNC was dissolved in 1 ml chloroform and then 1 ml oleylamine (OLA) was added. After 24 h, the decomposition was stopped by adding 3 ml of cyclohexane, and the nanoparticles were collected after a washing procedure, as described in the following section. To increase the size, instead of chloroform the same amount of pure OLA was used, or dimethylformamide was added. By replacing OLA with butylamine agave-shaped nanoarchitechtures (Fig. 1c) were obtained. Butylamine/dimethylaminopyridine resulted in houseleek shape and using 33% methylamine/ethanol gave coral fungi shape (Fig. 1c). To get the beta phase with the shape of starflower (Fig. 3a, *top-right*), 10 mg of tBOC-QNC was dissolved in 2.5 ml cyclohexanone and then 2.5 ml butylamine was added. After 44 h the decomposition was quenched. By using 5-aminotetrazole (0.5 g) as a coordination ligand for both sides of QA molecules, with 3 ml of 5 mg/ml chloroform solution of tBOC-QNC chrysanthemum shaped (Fig. 3a, *bottom-right*) micronanostructures were obtained after 1 month of decomposition. Supplementary Table 1 summarises the reaction conditions for the different crystals. After synthesis, the organic pigment micronanocrystals were isolated by adding cyclohexane in a volume ratio of 3:1 to the crude colloidal solutions, followed by centrifugation (relative centrifugal force = 14.100*g*, 5 min) and redispersion in chloroform. The washing step was repeated four times before the micronanocrystals were stored in chloroform or in chlorobenzene. Instead of centrifugation, sedimentation for more than 1 h followed by decantation can be applied, yielding the same results.

**Electrophysiological recordings**. Details on cell culture, viability and SEM measurements of biological samples can be found in the Supplementary Methods. Untransfected RBL cells were used for the recording of the K$^+$ inward rectifier. In these experiments the intracellular pipette solution contained 145 mM KCl, 1 mM MgCl$_2$, 10 mM Hepes, 10 mM glucose (pH 7.2) and the extracellular solution

contained 140 mM NaCl, 5 mM KCl, 2 mM CaCl$_2$, 1 mM MgCl$_2$, 10 mM Hepes, 10 mM glucose. Voltage ramps between −120 and +60 mV lasting for 100 ms (for long photostimulation in Fig. 7b) or 2 s (for short photostimulation in Fig. 7a) were applied from a holding potential of 0 mV. HEK 293 cells were transfected with 1 μg of YFP-TRPV1 DNA and 2 μl Transfectin reagent (Biorad). Electrophysiological experiments were performed 24–34 h after transfection using the patch-clamp technique in whole-cell recording configurations at 21 °C to 25 °C. An Ag/AgCl electrode was used as reference. For the study of the TRPV1 channel, voltage ramps were applied from a holding potential of 30 mV, covering a range of −100 to 100 mV over 2 s. The internal pipette solution included 145 mM cesium methanesulfonate, 20 mM EGTA, 10 mM Hepes, 8 mM NaCl, 3.5 mM MgCl$_2$ (pH 7.2). Standard extracellular solution consisted of 145 mM NaCl, 10 mM Hepes, 10 mM glucose, 5 mM KCl, 1 mM MgCl$_2$, 0.3 mM CaCl$_2$ (TRPV1), pH 7.4. Based on TRPV1 selectivity, inward currents of Na$^+$ and Ca$^{2+}$ are expected.

**Data availability**. The authors declare that the data supporting the findings of this study are available within the paper and its Supplementary Information files.

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

## Acknowledgements

We acknowledge financial support from the Austrian Science Fund FWF via the projects TRP 294-N19, FWF P28167-N34, and the Wittgenstein Prize for N.S.S.; as well as FWF Projects P26067 and P28701 to R.S. Support to E.D.G. from the Knut and Alice Wallenberg Foundation within the framework of the Wallenberg Centre for Molecular Medicine, Linköping University, is gratefully acknowledged. NMR spectrometers were acquired in collaboration with the University of South Bohemia (CZ) with financial support from the European Union through the EFRE INTERREG IV ETC-AT-CZ program (project M00146, 'RERI-uasb'). A part of the work was supported through the *Aufbruch Bayern* initiative of the state of Bavaria, and by the Czech Ministry of Education, Youth and Sports within the project Czech-BioImaging—LM2015062, and the Biological Chemistry cross-border Linz-České Budějovice study program.

## Author contributions

W.H. and E.D.G. conceived the project. M.S. developed and carried out the syntheses of hierarchical nanocrystals, analytics and optical characterisation. M.S. and O.M. carried out the syntheses and purification of precursors. M.J. prepared all samples for biological experiments and imaging studies. M.S., M.J. and J.N. did all SEM imaging. M.J., M.L. and R.S. did the cell culture and electrophysiology experiments. D.K., M.S. and J.S. made XRD measurements and analysis. W.S. carried out NMR measurements. M.S. and F.W.F. did modelling and calculations. N.S.S., R.S., W.H. and E.D.G. supervised and coordinated the work. E.D.G. wrote the manuscript with input from all authors.

## Additional information

**Competing interests:** The authors declare no competing financial interests.

