## [Peer Review File · Nature Communications]

Reviewers' comments:

Reviewer #1 (Remarks to the Author):

Sytnyk et al reported a new method of creating organic nano/micro-structured semiconductors based on a carbamate ester migration reaction. During the materials preparation, they tuned several reaction parameters (solvent, amine, etc.) and characterized different forms (morphology, polymorph, etc.) of the products. They also performed mechanistic studies to illustrate the growth process. Finally, the authors interfaced the materials with cells and performed photo-stimulations of the cells. This referee suggests a major revision.

1) In the introduction part, the authors mentioned that there was a lack of synthetic method to create hierarchical organic crystals. In fact, similar structures with the current one in the manuscript have already been reported and reviewed previously (e.g. *J. Mater. Chem.*, 2012, 22, 785). Other forms of nano/micro-structured organics were also reported before (e.g. *PNAS*, 2012, 109, 9287, *ACS Nano*, 2015, 10, 9486).

2) In the photo-stimulation part, the authors' claim about the stimulation mechanism needs to be revisited. Specifically, the authors' argument that the time scales of their observed photo effect (i.e. hundreds of milliseconds) were consistent with previously reported photothermal effects is not correct. In Ref. 45, Ghezzi et al used the photoelectric effect of semiconducting polymers to stimulate retinas using 20 ms light pulses. In Ref. 47, Martino et al did use the photothermal effect of semiconducting polymers but their results were opposite from what the authors observed in the manuscript. In short, 20 ms pulses can depolarize cell membranes but 200 ms pulses would cause hyperpolarization of cell membranes. In Ref. 48, Jiang et al used the photothermal effect of Si nanostructures but their stimulation conditions used 1 to 2 ms of pulses. Besides, their stimulation effect was based on a fast temperature increase induced capacitive current mechanism, similar to that in Ref. 46. In Refs. 49 and 50, the authors were using the photoelectric effect of Si, similar to that in Ref. 47. The authors of this manuscript are advised to measure the temperature profile following light illumination to prove the possible photothermal effect. The authors should also clearly categorize existing photo-stimulation mechanisms and clarify the one involved in this manuscript.

3) Other points related to the photo-stimulation part. In line 286, the authors said thin film QNC did not have the stimulation effect. This claim is not solid since the total mass of QNC in the thin film could be much less than the nanostructured microspheres. It is more likely that it requires longer light durations or higher light intensities to stimulate cells. In line 290, the authors claimed their work to be the first of such to use semiconductor photo-stimulation to modulate specific ion channel behavior. However, the authors did not show enough evidence that the photo-stimulation effect as observed in the manuscript was ion channel specific. In fact, the data presented showed that the stimulation was likely caused by membrane depolarization (involving multiple K^+ , Ca^{2+} channels) through TRPV channel activation. It may rather be a global effect of the cell membrane instead of an ion channel specific event as claimed by the authors. In addition, TRPV channel activation caused by photothermal heating of organic semiconductor has been reported already (e.g. *JACS*, 2016, 138, 9049).

Minor points:

1) In Fig. 1b, it is not clear what are the preparation conditions for structures I, II and III.

2) The authors also claimed that nanostructured quinacridone (QNC) has higher adhesion to cells versus smooth surfaces but the conclusion was drawn from a comparison between QNC microspheres and QNC thin films. Other factors, e.g. shape of QNC could also lead to the observed phenomenon. A more fair comparison would be to compare QNC nanostructured microspheres and QNC smooth microspheres. In addition, the authors showed higher cell densities on spheres with 10 μm in diameter versus 3 μm and 18 μm counterparts. They speculated that cells tend to grown on particles with similar sizes and shapes of themselves. This speculation needs more evidence, as the authors did not correlate the sizes of cells and particles in their data. In addition, other factors such as surface

roughness, porosity, curvature could all or partially contribute to the effect. Finally, in Fig. S15, it is not clear what the blue and green colors stand for.

3) The authors mentioned the nontoxicity of QNC for several times without experimental data supporting this argument. The authors are advised to perform standard cytotoxicity assays to verify the claim.

4) In Fig. 4, cross-sectional TEM would be more informative for the characterizations of the nano-bio interfaces.

5) In Fig. 5, what was the light spot size for all the stimulation? In Fig. 5b, n should be higher in the control group and p-value is needed for the comparison.

Reviewer #2 (Remarks to the Author):

Review

Cellular interfaces with biomimetic hydrogen-bonded organic semiconductor hierarchical nanocrystals
Sytnyk et al

The manuscript describes a body of work surrounding the synthesis of OSC nanocrystals and eventually their use for interfacing with live cells. The authors show remarkable control over the shape of the nanocrystals with innovative chemical synthesis.

Summary

The present manuscript describes very nicely the synthesis and characterization of OSC nanocrystals. The chemistry work is very detailed and the ability to make crystals of different shape under easy to use fabrication conditions is remarkable. The SEM images of the crystal structure are very beautiful. The weakness in the manuscript is related to the motivation and justification of the crystals for biological applications. Although the experiments shown are convincing with respect to interaction of the crystals with live cells, the utility of this method remains unclear. The interactions with the cells is not shown until Figure 4, with the previous 3 figures devoted to synthesis and characterization which, while very interesting, is not the main premise of the paper as indicated by the title. The very nice ability to tune the nanocrystal shape and size seems to be a central tenet of the paper, however this tunability has not been shown to demonstrate significant differences in terms of interfacing. For example if it could be shown that daggers versus needles resulted in different abilities to penetrate cells and effect changes in the cell, then the tunability becomes relevant to the biology. Some progress towards this is shown with respect to the size of the particles, however the data is not clear or convincing in this respect (SF 15). Another very important point is to demonstrate whether or not the needles/daggers are actually penetrating the plasma membrane, which is not clear from the ms. Finally the functionality of the OSC nanocrystals is described by the authors as being for photostimulation – however, photostimulation as a concept is not described until near the end of the paper.

Specific comments:

The correlation of the nanocrystals with green algae “the growth resembles that of green algae” – text from the abstract, appears to come out of nowhere, and while the images shown are beautiful and the similarity is interesting, there doesn’t appear to be any relevance connected with the function. This is an observation that right now seems irrelevant for the manuscript. There should also be a scale bar in the images of the algae shown in Fig.2 – in biological systems, morphological features may alter function depending on the length scale – ie. Nanoscale features may have orthogonal effects to micron, or mm scale features.

On line 140 the authors refer to the dendritic growth in the case of the green algae to maximize surface area, but again the parallel with the nanocrystal seems non-existent

In the intro line 45 the authors list examples for interrogation of single cells, however they should

specify that the examples are for intracellular events

The synthesis part is very interesting, and the deprotection reaction allowing carbamate esters to migrate between amine groups really appears novel. This begs the question whether this work alone would make a more coherent manuscript...

Figure 1. In part b, parts I and II should be referred to in the legend

Figure 3 shows again a very refined control of crystal structure. It seems to this reviewer that maximum impact in terms of biology would be achieved by the ability to show different function of the different crystals – for example by being able to use light of different wavelengths, or – again, by showing the utility of daggers versus needles for different applications.

On line 195 the authors describe conducting polymers and their application in life sciences. However what is misleading is that the authors describe properties such as low Young's modulus, while the nanocrystals appear to be stiff, rigid structures (which is described as an advantage for penetrating the cell). In fact what appears to give most conducting polymers good mechanical properties with respect to biology, is the uptake of ionic solutions rendering them hydrogel like.

Line 209: the authors say the crystals are cell culture scaffolds, however this has quite distinct meaning in tissue engineering and should be altered, perhaps to substrate

Line 215 – the authors state that the PM invaginates around the nanoneedle structures, however it seems more like an engulfment type mechanism. This also begs the question are the needles penetrating the cell? If so, viability studies are necessary.

Line 221 – hedgehogs are drop-cast onto a glass slide – do they stay and adhere or is the sublimation of the film necessary to maintain them in position?

The interaction with cells themselves is very interesting, however it is worth mentioning that typically cells such as HEK should adhere to surfaces -when cells are rounded up as shown in the images, it often means that they are trying to avoid contact with the surface.

In general, the details for cell culture are inadequate. The culture conditions and the time of interaction with the hedgehogs is necessary. How are the QNC sterilized?

SI fig 15 – the legend is not adequate to describe the images. What is the difference between left and right. To me there does not appear to be a significant difference. What do the error bars represent?

Figure 4. The statement 'overwhelming preference' for adhering to nanostructures should be quantified.

The fact that the RBL cells can rip nanoneedles out – what does this imply for function?

Line 286 – the control should be non-transfected HEK cells, and not RBL cells.

The statement "photothermal heating effects are useful in both in vitro and in vivo applications" should be qualified - if it is so important it should be in the introduction. In fact, there is a general need to be a lot more explicit about how this technique could be useful, and of course how it would replace optogenetics, if that is what the authors are implying in line 289

Line 319 on – the use of these materials for in vivo electrophysiology would rely on their conductivity. How conductive are these materials? How would their use be envisioned – for sensing or stimulation?

Line 324 – a tattoo is not an implant

Reviewer comments: RED

Our response: BLUE

Text added to revised manuscript: "BLACK"

Reviewer #1:

Reviewer #1 (Remarks to the Author):

Sytnyk et al reported a new method of creating organic nano/micro-structured semiconductors based on a carbamate ester migration reaction. During the materials preparation, they tuned several reaction parameters (solvent, amine, etc.) and characterized different forms (morphology, polymorph, etc.) of the products. They also performed mechanistic studies to illustrate the growth process. Finally, the authors interfaced the materials with cells and performed photo-stimulations of the cells. This referee suggests a major revision.

1) In the introduction part, the authors mentioned that there was a lack of synthetic method to create hierarchical organic crystals. In fact, similar structures with the current one in the manuscript have already been reported and reviewed previously (e.g. J. Mater. Chem., 2012, 22, 785). Other forms of nano/micro-structured organics were also reported before (e.g. PNAS, 2012, 109, 9287, ACS Nano, 2015, 10, 9486).

Thank you for pointing out this misstep and indicating a set of very pertinent references. We have now corrected this misleading inaccuracy to properly refer to the previous work. What we meant to express is that **colloidal** or **ligand-mediated** routes for organics have not been reported before. We would point out that in these published papers there is no mention of "ligands" or "colloids". Ligand-mediated approaches which are a very mature discipline for inorganic materials, have not been followed up in the field of organic crystals – and we show a novel way how this can be done. We have added to the text the following to outline our motivation and frame it in the context of both inorganic and organic crystalline materials:

Page 3... "We have chosen a ligand-mediated approach not only to afford fine synthetic control of the structure, but to yield a crystal surface modified with a ligand monolayer suitable for favourable interaction with lipid bilayer cell membranes. Nanocrystalline architectures of organics can be grown using various vapour or solution deposition/evaporation techniques²⁶⁻²⁸. Organic analogues of the colloidal syntheses^{29,30}, however, so prevalent in the case of hierarchical inorganic materials³¹⁻³⁵ have not been reported. We have recently introduced the idea of ligand-mediated syntheses of colloidal organic monocrystals³⁶. To form hierarchical assemblies, herein we introduce a different chemical route, involving ligand-mediated precursor decomposition at room temperature followed by nucleation and assembly into hierarchical structures."

2) In the photo-stimulation part, the authors' claim about the stimulation mechanism needs to be revisited. Specifically, the authors' argument that the time scales of their observed photo effect (i.e. hundreds of milliseconds) were consistent with previously reported photothermal effects is not correct. In Ref. 45, Ghezzi et al used the photoelectric effect of semiconducting polymers to stimulate retinas using 20 ms light pulses. In Ref. 47, Martino et al did use the photothermal effect of semiconducting polymers but their results were opposite from what the authors observed in the manuscript. In short, 20 ms pulses can depolarize cell membranes but 200 ms pulses would cause hyperpolarization of cell membranes. In Ref. 48, Jiang et al used the photothermal effect of Si nanostructures but their stimulation conditions used 1 to 2 ms of pulses. Besides, their stimulation effect was based on a fast temperature increase induced capacitive current mechanism, similar to that in Ref. 46.

We are grateful for these useful and insightful critiques - we have now revisited the mechanism according to these comments. To accomplish this, we carried out a full set of new experiments using faster time scale of illumination in-line with the recent findings exploring the topic of

nano/micromaterial-mediated cellular photostimulation. We used pulses of 10 ms – 50 ms, similar as used by Ghezzi (*Nat. Photonics* **7**, 400–406, 2013) and Martino (*Sci. Rep.* **5**, 8911, 2015) and with total energy doses similar to those reported recently by Jiang *et al.* (*Nature Mat.* **15**, 1023, 2016). By choosing these ranges we were able to clearly see two phenomena happening at different time scales - rapid depolarizing current transients (which we had previously not observed when using long CW-illumination and which are quite similar to what Jiang *et al.* have recently reported) occurring at the beginning of the light pulse as well as sustained increased current occurring at longer time-scales. This slower component is correlated to ion currents through specific types of channels, and the channel current-voltage characteristics do not change, only the magnitude of current flowing while the channel is open, which strongly implicates a thermal mechanism. This reversible effect is now visible both in the new pulsed-illumination study we have done and the previous CW-illumination measurements. The consistency of this finding over different timescales evidences the photothermal interpretation. Our work corroborates the rapid thermocapacitive effect implicated in recent studies on excitable cells and shows how this is manifested in ion channel current measurements. We believe our findings showing this reversible effect on top of the faster depolarizing current behavior is a novel and useful advance to this field and to the ongoing discussion in the literature. Moreover, we would argue that by using voltage-clamp measurements and measuring rectifying channels we have proper verification that intense and fast heating of nanoparticles doesn't lead to nonspecific cell membrane leakage. We kindly refer the reader to the *Photostimulation of ion channels* section of the manuscript for detailed text, which has been updated with respect to the original version. For convenience, we include here the updated Figure 7:

Figure 7 | Hedgehog-mediated photostimulation of ion channels. a, Reversible photostimulation of the K^+ inward rectifier channel with 10 ms light pulses in an RBL cell grown on a $10\ \mu\text{m}$ -size nanoneedle hedgehog apparent in i-v measurements in whole-cell configuration. Rapid ($<1\ \text{ms}$ time scale) light-induced depolarizing currents are visible over the entire voltage sweep range, however only where the K^+ inward rectifier channel is open ($V < -80\ \text{mV}$) are there photoinduced increases ($+30\%$) in K^+ inward current. The inset shows the effects of light pulses, at a holding potential of $-110\ \text{mV}$ with the depolarizing current spikes and a plateau region of K^+ current increase, the latter attributed to local photothermal heating and faster ion diffusion. b, The K^+ rectifier in an RBL cell on hedgehog measured using cw-

illumination over 1s, 5 mW/mm². The i-v characteristic remains the same, only the peak value of K⁺ current increases reversibly with illumination. This is consistent with photothermal heating raising the local temperature of the cell, increasing the current of ions through the open channels. The inset shows statistics for photoinduced K⁺ influx increase percentage (K⁺ rectifier) of the illuminated RBL cells growing on nanoneedle- and nanodagger-type hedgehogs. Error bars represent the standard error of the mean (n=11-14). The photomicrograph shows how cells in contact with hedgehogs appear under the microscope. c, Current-voltage relationship of cation current in temperature-sensitive TRPV1 channels overexpressed in HEK cells on hedgehogs. Depolarizing current spikes are visible throughout the whole trace while increases in cation current are apparent in voltage ranges where the channel is open. d, HEK cell with TRPV1 give rapid and reversible photoinduced stimulation of cation influx when measured at the cell resting potential, -60 mV. d, Schematic illustration of the mechanisms possible at nanocrystal/cell interfaces. Photocapacitive membrane depolarization can occur on a short time-scale (left) and rapid photothermal changes can lead to capacitive modulation at short time scales, while increases in through-channel conductance due to localized photothermal heating (right) can be observed on time scales from 10 ms up to 1000 ms.

In Refs. 49 and 50, the authors were using the photoelectric effect of Si, similar to that in Ref. 47. The authors of this manuscript are advised to measure the temperature profile following light illumination to prove the possible photothermal effect. The authors should also clearly categorize existing photo-stimulation mechanisms and clarify the one involved in this manuscript.

Thanks for this comment, we have strived in the revision to make clear distinctions between the different photothermal mechanisms and frame them in the context of the recent literature on this topic (please see new *Photostimulation* section). We have now done modeling and simulation to help understand photoinduced heating in hedgehog structures. There is a new SI Figure S15 showing these results. The new Figure 7e (shown above) details the different mechanisms.

SI Figure 15. | Photothermal heating of different QNC micro-objects. Heating simulations under laser beam were performed with COMSOL Multiphysics 4.2a software (COMSOL Inc.). Heating of different micro objects was modeled by the differential equation for heat transfer in solids by assuming that the radiation energy was absorbed by the surface. The laser power was set as CW 20 mW/cm² over 1 second. The heat dissipation was modeled with convective heat flux to the water environment with heat transfer coefficient of 40 W/(m²K). Hedgehog samples heat up to higher temperature faster than either spheres or planar layers.

3) Other points related to the photo-stimulation part. In line 286, the authors said thin film QNC did not have the stimulation effect. This claim is not solid since the total mass of QNC in the thin

film could be much less than the nanostructured microspheres. It is more likely that it requires longer light durations or higher light intensities to stimulate cells.

Thanks for pointing this out, this comparison is indeed not relevant. We have removed this discussion as there are many complications with comparing QNC hedgehogs to planar films: first, the films are not truly planar, they are nanostructured to various degrees as well, and, critically, HEK cells do not grow on these QNC films at all, so no fair comparison is possible for HEK cells. The heating effect of films vs. larger microstructures can be understood better after modeling the temperature increase during illumination, and the dynamics of heat dissipation, as we have shown in the new SI Figure S15.

In line 290, the authors claimed their work to be the first of such to use semiconductor photo-stimulation to modulate specific ion channel behavior. However, the authors did not show enough evidence that the photo-stimulation effect as observed in the manuscript was ion channel specific. In fact, the data presented showed that the stimulation was likely caused by membrane depolarization (involving multiple K⁺, Ca²⁺ channels) through TRPV channel activation. It may rather be a global effect of the cell membrane instead of an ion channel specific event as claimed by the authors. In addition, TRPV channel activation caused by photothermal heating of organic semiconductor has been reported already (e.g. JACS, 2016, 138, 9049).

Thanks for these points. As now elaborated in the text, we have found that the rapid depolarizing current is certainly involving multiple ions, however the conditions we use for current-voltage sweeps are specifically measuring behavior of the K⁺ rectifier and TRPV1 channels, respectively. The measurements we have done are ion channel-specific in that we are utilizing measurement protocols to measure specific types of channels selectively – measurement sweep conditions, electrolyte, and most importantly in the case of HEK cells the overexpression of TRPV1. Moreover, TRPV1 channels are known to be highly Ca²⁺ selective. The current-voltage scans show characteristic behavior corresponding to the given channels, a characteristic which is not changing with sustained illumination and not altering gating behavior, proving that 1) the fast depolarizing current (yes this is a ‘global’ effect) is a passive capacitive effect which doesn’t lead to modulation of these ion channels, the perturbation is not large enough to cause this in the case of the channels we measured, and 2) a reversible local thermal heating leads to higher current through the already-open channels. This second conclusion is valid because of the ion channel specificity of the measurement we are conducting. Our point is that the other reported studies of photostimulation have not evaluated the electrophysiology of channels in this way, and the information we have obtained is insightful. The closest relevant previous report, and fitting with our findings, is the recent JACS paper (we had not been aware of previously, we are grateful to the reviewer for pointing it out to us!). This interesting calcium-imaging study corroborates the generalized photothermal heating mechanism implicated in our findings. We have now cited and discussed this work in context. We have revised our text accordingly and kindly refer to the rewritten *Photostimulation* section.

Minor points:

- 1) In Fig. 1b, it is not clear what are the preparation conditions for structures I, II and III.

We have now added to the caption that the size depends on precursor starting concentration.

2) The authors also claimed that nanostructured quinacridone (QNC) has higher adhesion to cells versus smooth surfaces but the conclusion was drawn from a comparison between QNC microspheres and QNC thin films. Other factors, e.g. shape of QNC could also lead to the observed phenomenon. A more fair comparison would be to compare QNC nanostructured microspheres and QNC smooth microspheres. In addition, the authors showed higher cell

densities on spheres with 10 μm in diameter versus 3 μm and 18 μm counterparts. They speculated that cells tend to grow on particles with similar sizes and shapes of themselves. This speculation needs more evidence, as the authors did not correlate the sizes of cells and particles in their data. In addition, other factors such as surface roughness, porosity, curvature could all or partially contribute to the effect. Finally, in Fig. S15, it is not clear what the blue and green colors stand for.

This is a good idea! Despite sustained attempts, we have not able to synthesize “flat” spherical QNC colloidal crystals reproducibly. Yes, this would be a more fair comparison between hedgehog structures and “flat” QNC, and we are currently working on mastering methods for colloidal control to achieve such shapes. As to the issue with the different particle sizes, we have removed this figure and discussion as we agree it is too preliminary to draw useful conclusions.

3) The authors mentioned the nontoxicity of QNC for several times without experimental data supporting this argument. The authors are advised to perform standard cytotoxicity assays to verify the claim.

Thanks for this point, we agree that this was a major deficiency, we have now corrected this:

CytoTox-Glo™ cytotoxicity assay from Promega was performed according to the producer protocol. Hedgehog crystals in CHCl_3 and QNC powder dispersed in EtOH were deposited directly into a 96-well flat-bottom polypropylene culture plate. The QNC thin film was evaporated on a PDMS-coated PET foil, cut, and glued to the bottom of the wells. The total mass of QNC was kept the same for all experimental conditions. The plates were then UV sterilized and seeded with 2000 cells. The cytotoxicity was determined after 4, 24, 48 and 72h without culture media exchange during the whole experiment.

SI Figure 13 | The viability of HEK and RBL cells is not adversely affected by QNC in the form of evaporated thin films, ground powders, or hedgehogs. The CytoTox-Glo™ luminescence-based assay was used (Promega) to allow the viability assays to be carried out and measured directly in cell-culture plates. The last-day data points show increased cell death, due to overgrowth of cells and depletion of nutrients from the culture media, which was not exchanged over the course of the experiment. (n = 9). For HEK cells (left), control showed slightly higher cytotoxicity than experimental samples, however none of the differences were statistically significant, $p < 0.05$ (one-way ANOVA).

Moreover, we have done detailed SEM imaging of the cells growing on hedgehogs as a function of their culture time, and based on the progression of spreading and cell proliferation we likewise infer good viability of cells on these nanostructured materials.

4) In Fig. 4, cross-sectional TEM would be more informative for the characterizations of the nano-bio interfaces.

We are grateful for this constructive point, indeed this is important since we are trying to establish high surface-area contacts with minimal semiconductor/cell cleft. Unfortunately we could not obtain access for suitable TEM, however we have been successful to complement our more detailed SEM studies now with SEM cross-sections (in addition to new imaging studies showing the progression of the interface as a function of cell culture time). We kindly refer the reviewers to the section *Interfaces with single cells* for the updated text on page 11 and 12. The cross-sectional SEM has become the new Figure 6:

Figure 6 | Hierarchical hedgehogs crumple, buckle, and rearrange under the growing cell.
a, An interface between a HEK cell and nanodagger hedgehog, with close contacts between cell

membrane and nanocrystallites labelled and shown in higher magnification. **b**, Crosssection overview of another HEK/nanodagger interface, clearly showing the extent of distortion of the nanodagger assembly under the cell. Scale bar = 500 nm.

5) In Fig. 5, what was the light spot size for all the stimulation? In Fig. 5b, n should be higher in the control group and p-value is needed for the comparison.

We have now clarified in the beginning of the *Photostimulation* section the spot size, ~15 μm diameter. We have now included the n values for number of measured cells, in the main text and in figure captions, where appropriate, and also p-value (SI Figure S13).

Reviewer #2 (Remarks to the Author):

The manuscript describes a body of work surrounding the synthesis of OSC nanocrystals and eventually their use for interfacing with live cells. The authors show remarkable control over the shape of the nanocrystals with innovative chemical synthesis.

Summary

The present manuscript describes very nicely the synthesis and characterization of OSC nanocrystals. The chemistry work is very detailed and the ability to make crystals of different shape under easy to use fabrication conditions is remarkable. The SEM images of the crystal structure are very beautiful. The weakness in the manuscript is related to the motivation and justification of the crystals for biological applications. Although the experiments shown are convincing with respect to interaction of the crystals with live cells, the utility of this method remains unclear. The interactions with the cells is not shown until Figure 4, with the previous 3 figures devoted to synthesis and characterization which, while very interesting, is not the main premise of the paper as indicated by the title. The very nice ability to tune the nanocrystal shape and size seems to be a central tenet of the paper, however this tunability has not been shown to demonstrate significant differences in terms of interfacing. For example if it could be shown that daggers versus needles resulted in different abilities to penetrate cells and effect changes in the cell, then the tunability becomes relevant to the biology. Some progress towards this is shown with respect to the size of the particles, however the data is not clear or convincing in this respect (SF 15). Another very important point is to demonstrate whether or not the needles/daggers are actually penetrating the plasma membrane, which is not clear from the ms. Finally the functionality of the OSC nanocrystals is described by the authors as being for photostimulation – however, photostimulation as a concept is not described until near the end of the paper.

We are grateful for the reviewer for these comments, we agree that we had not outlined our motivations sufficiently or justified the steps we took in our materials choice and development (please see the new *introduction* section). We explain now from the beginning the significance of how we have designed our materials to form favorable interfaces and how these choices and the results fit into the context of others who endeavor to do this with inorganic materials. The field of cellular photostimulation is also brought into focus from the outset of the paper. Accordingly, we have significantly overhauled the introduction text as well as the “Interfaces with single cells” and “photostimulation of ion channels” sections, and we believe we have fixed these shortcomings. With detailed cell culture studies and evaluation of the interface formation over time, as well as strong evidence from cross-sections that cell membrane penetration does not occur, we would argue that these concerns are largely addressed.

Specific comments:

The correlation of the nanocrystals with green algae “the growth resembles that of green algae” – text from the abstract, appears to come out of nowhere, and while the images shown are beautiful and the similarity is interesting, there doesn’t appear to be any relevance connected with the function. This is an observation that right now seems irrelevant for the manuscript. There should also be a scale bar in the images of the algae shown in Fig.2 – in biological systems, morphological features may alter function depending on the length scale – ie. Nanoscale features may have orthogonal effects to micron, or mm scale features. On line 140 the authors refer to the dendritic growth in the case of the green algae to maximize surface area, but again the parallel with the nanocrystal seems non-existent.

Thank you for pointing out this weakness. We agree that this conceptual parallel is, from a scientific point of view, insufficiently supported and not especially relevant to the synthetic or biointerface message of the manuscript. The mechanism and similarities would need more evaluation to be significant, therefore in line with this comment we remove the algae comparison. We reformatted Figure 2 to show the detail of the time-dependent growth:

Figure 2 | SEM imaging of QNC colloidal crystal aliquots removed at different times during synthesis reveals the growth process of hierarchical crystals. Branching is occurring already at 15 min, and continues as the microstructures grow over 68h. Monomeric QNC molecules precipitate onto exposed facets of the crystals, generating growth along this facet that progressively ceases to grow due to increased steric hindrance caused by attached alkylamine ligands. Repetition of this process maintains branched growth as the microstructures expand spherically.

In the intro line 45 the authors list examples for interrogation of single cells, however they should specify that the examples are for intracellular events.

We have now clearly made this distinction in the text, the introduction has been significantly rewritten.

The synthesis part is very interesting, and the deprotection reaction allowing carbamate esters to migrate between amine groups really appears novel. This begs the question whether this work alone would make a more coherent manuscript...

It is very nice to receive positive appreciation for the novelty of the synthetic approach, and fully agree that in the originally-submitted manuscript the motivation and connection to the biological experiments was far from optimal. We believe that the rewritten introduction and more detailed cell interface and viability experiments now make a coherent and logical story. We also argue that this type of approach to new materials should be pursued in the biological applications context.

Figure 1. In part b, parts I and II should be referred to in the legend.

We have fixed this now:

“b, The size of hedgehog crystals is determined by the starting concentration of tBOC-QNC, allowing narrow size distribution to be achieved. Three sizes - I, II, and III are shown in the SEM micrographs.”

Figure 3 shows again a very refined control of crystal structure . It seems to this reviewer that maximum impact in terms of biology would be achieved by the ability to show different function of the different crystals – for example by being able to use light of different wavelengths, or – again, by showing the utility of daggers versus needles for different applications. On line 195 the authors describe conducting polymers and their application in life sciences. However what is misleading is that the authors describe properties such as low Young’s modulus, while the nanocrystals appear to be stiff, rigid structures (which is described as an advantage for penetrating the cell). In fact what appears to give most conducting polymers good mechanical properties with respect to biology, is the uptake of ionic solutions rendering them hydrogel like.

We thank you for this critique, indeed the initial parallel we made was misleading. We would rather argue that we are offering a new materials class which though organic, features traits in terms of structurizability and mechanical robustness that make them resemble more silicon. We have rewritten the section *Interfaces with single cells*, starting on page 11 and have reframed our results and also supplemented this with new findings from cell-culture-over-time experiments and SEMs of cell/semiconductor interface cross-sections.

“Page 13... This leads us to conclude a unique mechanical advantage - the hierarchical structure has the mechanical freedom to move, and the constituent nanocrystallites can be rearranged by the growing cell. This interpretation is consistent with the observation discussed previously of RBL cells ripping nanocrystallites out of the hierarchical parent structures. This behaviour can easily be rationalized considering that the QNC molecules in the nanocrystals are held together by a strong interplay of π - π stacking and hydrogen-bonding, while the hierarchical arrangements are held together by much weaker van der Waals forces. These findings offer a new paradigm for organic semiconductors for bioelectronics – hierarchical architectures can show mechanical plasticity while being constituted of a rigid and stable material.”

Line 209: the authors say the crystals are cell culture scaffolds, however this has quite distinct meaning in tissue engineering and should be altered, perhaps to substrate

We have now removed this misleading terminology – it’s true we are not going in the direction of any tissue culture, though we now discuss in the conclusions and outlook about possible further research directions.

Line 215 – the authors state that the PM invaginates around the nanoneedle structures, however it seems more like an engulfment type mechanism. This also begs the question are the needles penetrating the cell? If so, viability studies are necessary.

We have now specified that primarily invagination is observed, though some endocytosis-like events can be seen in the case of HEK cells. We have addressed the critical question of plasma membrane penetration by both carrying out a standard viability protocol over the course of cell culture time and by measuring cell/hedgehog cross-sections.

Line 221 – hedgehogs are drop-cast onto a glass slide – do they stay and adhere or is the sublimation of the film necessary to maintain them in position?

We have now clarified this point in the text more thoroughly. The hedgehogs stay put after drop casting, evaporated films were only used for comparison where indicated, otherwise no other samples featured evaporated layers and they are not needed for any adhesion purposes:

“From the outset of this study, we surmised that the organic needle-like hierarchical QNC hedgehogs should be capable of forming close and high-surface area contacts with cells. To evaluate this, we first drop-cast colloidal solutions of QNC hedgehogs on glass substrates suitable for cell culture. The two types of γ hedgehogs shown in Figure 3a were used: hierarchically-assembled arrays of nanoneedles, while the second is the nanodagger modification, featuring nano-scale triangular daggers tapering down to a sharp 10-20 nm wide point. Drop-cast hedgehogs were found to adhere well to both hydrophobic (OTS-modified) as well as hydrophilic ((3-Aminopropyl)triethoxysilane, APTES-modified) glass substrates, and did not delaminate in cell culture solutions, or after UV sterilization.”

The interaction with cells themselves is very interesting, however it is worth mentioning that typically cells such as HEK should adhere to surfaces -when cells are rounded up as shown in the images, it often means that they are trying to avoid contact with the surface.

We are grateful for this important critical point and question. We have now evaluated using SEM the evolution of cell shape and interface formation with the hedgehog crystals over cell culture time. In the cell culture time-dependence we see that over the course of many hours the HEK cell morphology changes and the cells spread on the hedgehog surfaces extensively. The rounded shape is apparent at shorter times in culture. We have added the following text and also two expanded SEM figures – one for RBL and the other for HEK cells. The progressive spreading of HEK cells is clearly visible over time.

“Cell culture was carried out for up to three days, with samples removed for SEM imaging and cell viability assays. Glass substrates modified to be favourable for the given cell culture (hydrophobic silanised glass for RBL cells and APTES-modified for HEK) were used for SEM imaging. The RBL cells were found to grow on the QNC planar films prepared by vacuum sublimation, however HEK cells showed no attachment. On the other hand, we have found that RBL and HEK cells readily attach to both types of hedgehogs. RBL cells form remarkably conformable interfaces with the nanostructured surface of hedgehog crystals already after 1.5h in culture, apparent from SEM (Figure 4). The cell/crystal interfaces can be conveniently viewed also by optical microscopy, where the crystals’ luminescence allows fluorescence imaging of the semiconductor structures (SI Figure S11). From SEM imaging, it is apparent that the plasma membrane and extracellular matrix invaginates around the nanoneedle structures, causing anchoring of cells onto the microhedgehog surface (Figure 4). The evaluation of the true interface cleft distance *in vivo* is a question of current debate, as the fixation of cells for electron microscopy can produce a cleft morphology different from what is present *in vivo*^{15,54}. The cleft distance has been determined to change as much as 10-50 nm following fixation in some cases,

even where cryofixation methods are employed. In the case of SEM imaging of hedgehog/cell interfaces, the inevitable “artefact problem” of cell shrinkage occurring during fixation actually yields interesting information: We have observed frequent examples where the hedgehogs are actually pulled apart by cells, with cells extracting nanoneedles from the parent microstructure, or sometimes splitting the hedgehog structure between multiple cells (Figure 4c, SI Figure S12). This indicates that the membrane/nanoneedle interaction can be stronger than the forces holding the hierarchical architecture together. Over longer times in culture, the cells are found to transition from a rounded shape to spread more extensively on the hedgehogs (Figure 4d). This progressive behaviour is consistent with the interpretation that the cells are viable while having such an extensive contact area with the crystals, which is further corroborated using the CytoTox-Glo luminescence-based assay (Figure S13). Carried out over three days, the assay demonstrated no difference in viability between cells cultured with hedgehogs, planar or powder QNC, and control samples. This experiment suggests that neither form of the QNC material is acutely cytotoxic. QNC itself, as a commercial pigment, has been studied with regards to consumer safety and determined to be nontoxic⁴⁶. Like RBLs, HEKs form a high interfacial area contact with hedgehog crystals, with evidence of extensive invagination of the cell membrane around the sharp nanocrystallites (Figure 5), with occasional examples of engulfment-type processes¹⁵ clearly occurring (Figure 5b). As mentioned before, HEK cells do not attach to planar evaporated QNC films. To discriminate the role of hierarchical nanostructure versus QNC surfaces themselves, we prepared samples with hedgehogs drop-cast onto a glass slide, followed by sublimation of a uniform 80 nm thin film of QNC over the entire sample area, including on top of the hedgehogs. HEK cells were found to grow exclusively on the hedgehogs and not anywhere on the planar films (Figure 5c, SI Figure S14). From this it is clear that hedgehog structures promote cell attachment and growth due to their nano- microstructure³. HEK cells progressively spread over hedgehog structures during cell culture (Figure 5d), with uncompromised viability (SI Figure S13).”

Figure 4 | RBL cell culture with QNC nanohedgehogs. **a**, RBL cells cultured on 10 μm -size nanoneedle hedgehogs, with cells false-coloured pink. The image captures the cells when they are degranulating. **b**, Tight interfaces between RBL cells and nanoneedle hedgehogs. **c**, The cell membrane and extracellular matrix of RBL cells can rip nanoneedles out of the hierarchical microcrystal, demonstrating the strength of the cell/nanocrystals interaction. **d**, RBL cell culture interrupted at different intervals, showing the progression of cell attachment, proliferation, and morphology over time. Scale bars = 5 μm .

Figure 5 | HEK cell culture with QNC nanohedgehogs. **a**, Close and highly-conformal interfaces form after a few hours in culture. **b**, HEK cells do not grow on evaporated QNC thin-films. Here hedgehogs are deposited on a uniform thin-film, showing the overwhelming preference for hedgehogs over planar films. **c**, Interface contact between HEK cells and hedgehogs increases over cell culture time, and morphology transitions from rounded cells with a few attachment points to cells showing extensive spreading on the hedgehog structures. Scale bar = 2.5 μm

In general, the details for cell culture are inadequate. The culture conditions and the time of interaction with the hedgehogs is necessary. How are the QNC sterilized?

Thanks for pointing this out. We have now added the relevant info: The QNC is sterilized using UV light or ethanol without apparent detriment to the samples. This part is described in the main text, with further details now in the SI experimental methods.

“Hedgehog microcrystals were drop cast from chloroform onto glass slides coated with (3-Aminopropyl)triethoxysilane, APTES (Human embryonic kidney HEK cells) or a monolayer of n-octyltrichlorosilane, OTS (Rat basophilic leukemia RBL cells), which were used for cell seeding. These self-assembled monolayer modifications were done by first treating glass samples with oxygen plasma, then transferring them into a sealed glass chamber with an open vial of OTS or APTES, and then heating the chamber to 70°C, 3h for APTES and 90°C, 1h, for OTS. The samples were then rinsed with isopropanol and sonicated in isopropanol (APTES) or toluene (OTS) to remove physisorbed layers. Samples were then sterilized prior to cell culture UV light. Human embryonic kidney 293 (HEK) cells were cultured in DMEM supplemented with L-glutamine (2 mM), streptomycin (100 $\mu\text{g}/\text{ml}$), penicillin (100 U/ml), and 10% fetal calf serum, the rat basophilic leukemia 2H3 (RBL) cells were grown in MEM supplemented with 10% fetal calf serum, 2 mM glutamine, 2 U/ml penicillin and 2 mg/ml streptomycin, and incubated at 37°C, 95% humidity and 5% CO_2 .”

SI fig 15 – the legend is not adequate to describe the images. What is the difference between left and right. To me there does not appear to be a significant difference. What do the error bars represent?

We have removed this figure and discussion as we agree it is too preliminary to draw useful conclusions.

Figure 4. The statement ‘overwhelming preference’ for adhering to nanostructures should be quantified.

Together with cell growth studies on hedgehogs, we have conducted cell culture experiments on 80 and 320 nm-thick QNC films on glass. We have found that RBL cells can adhere to both evaporated films and hedgehogs and proliferate, however we were not able to observe any HEK cells to grow on these 80 or 320 nm-thick films, and we can conclude that their preferential growth on hedgehogs is due to morphology, since both hedgehogs and the adjacent substrate are covered with a uniform thickness of ‘chemically’ exactly the same quinacridone molecules:

“As mentioned before, HEK cells do not attach to planar evaporated QNC films. To discriminate the role of hierarchical nanostructure versus QNC surfaces themselves, we prepared samples with hedgehogs drop-cast onto a glass slide, followed by sublimation of a uniform 80 nm thin film of QNC over the entire sample area, including on top of the hedgehogs. HEK cells were found to grow exclusively on the hedgehogs and not anywhere on the planar films (Figure 5c, SI Figure S14). From this it is clear that hedgehog structures promote cell attachment and growth due to their nano- microstructure. HEK cells progressively spread over hedgehog structures during cell culture (Figure 5d), with uncompromised viability again evidenced using the CytoTox assay.”

The fact that the RBL cells can rip nanoneedles out – what does this imply for function?

We are grateful to the reviewer for this comment – it has encouraged us to pursue this question more deeply in the revision. Originally, the conclusion we could draw from this observation was that the cell membrane/nanocrystal interaction could be stronger than the forces which hold together the hierarchical assembly of nanocrystals. To strengthen this message, we have added a new SI Figure S12, showing more SEMs demonstrating this behavior. Evaluating the question more deeply and looking at cross-sectional SEMs, the picture that emerges is that the hierarchical crystals, though constituted of a rigid, crystalline material, are quite plastic. They can crumple and distort to accommodate a growing cell, which places them in an interesting category in terms of mechanical properties of materials for biointerfacing – they are rigid yet the hierarchical structure is accommodating. We have expanded this discussion arguing this as a useful feature that is a new paradigm for choosing materials for cellular interfaces:

“Page 12...In the case of SEM imaging of hedgehog/cell interfaces, the inevitable “artefact problem” of cell shrinkage occurring during fixation actually yields interesting information: We have observed frequent examples where the hedgehogs are actually pulled apart by cells, with cells extracting nanoneedles from the parent microstructure, or sometimes splitting the hedgehog structure between multiple cells (Figure 4c, SI Figure S12). This indicates that the membrane/nanoneedle interaction can be stronger than the forces holding the hierarchical architecture together.”

“page 13...the hierarchical nanocrystalline structure partially collapses, buckles, and distorts to give way under the cell. Even though the extent of the distortions revealed by SEM is a result of fixation stresses, we find no examples of nanocrystallites breaking the integrity of the cell membrane. This leads us to conclude a unique mechanical advantage - the hierarchical structure has the mechanical freedom to move, and the constituent nanocrystallites can be rearranged by the growing cell. This interpretation is consistent with the observation discussed previously of RBL cells ripping nanocrystallites out of the hierarchical parent structures. This behaviour can easily be rationalized considering that the QNC molecules in the nanocrystals are held together

by a strong interplay of π - π stacking and hydrogen-bonding, while the hierarchical arrangements are held together by much weaker van der Waals forces. These findings offer a new paradigm for organic semiconductors for bioelectronics – hierarchical architectures can show mechanical plasticity while being constituted of a rigid and stable material.”

SI Figure 12 | RBL cells deconstructing hedgehogs and rearranging nanocrystallites, shows that though the nanocrystals themselves are rigid, the hierarchical arrangement is plastic. Scale bar = 2 μ m.

Line 286 – the control should be non-transfected HEK cells, and not RBL cells.

We would like to point out that the RBL cells are never controls for HEKs. RBL “experimental” cells are those growing with contact to hedgehog structures, while controls are those growing on planar substrates, and both are irradiated with light and the endogenous K^+ rectifier channels are apart in the type of voltage sweep that we use. We have now rewritten this section in a more straightforward way and we separate the discussions on RBL cells, which is then followed with HEK cell experiments, we kindly direct the reviewers to the updated *Photostimulation* section.

The statement “photothermal heating effects are useful in both in vitro and in vivo applications” should be qualified - if it is so important it should be in the introduction. In fact, there is a general need to be a lot more explicit about how this technique could be useful, and of course how it would replace optogenetics, if that is what the authors are implying in line 289

We have endeavored in the introduction to make our motivation clearer and put the work in context, especially with respect to very recent advancements in this field occurring in the past few years. We have added the following to the introduction, and also kindly refer the reviewers to the new conclusion section:

Page 3...“Artificial high-aspect nanostructures have similarly been a focus of interest for electronic interfacing with living cells, being sought after for applications in high-quality extra- and intracellular electrophysiology^{7,15} recording and stimulation and for providing a bridge into the cytosol for both delivery and intracellular sensing^{10,11}. Inorganic materials, especially silicon,

and metals like platinum and gold predominate in all these applications. A common goal is getting as close to the cell as possible, forming a minimal cleft, and ideally with large interfacial area^{13,15}. Optimizing such structures is especially critical in the case of (opto)electronic interfaces, where the cleft between the cell and electronic element results in electric field screening and poor coupling¹⁶⁻¹⁸. Recently, close cellular interfaces with nanoscale amorphous silicon particles have been able to show reversible photostimulation of excitable cells¹⁹. Control of biology with light at the single-cell level is a concept with far-reaching consequences in both fundamental biological research and applied medicine. Optogenetics is widely considered to be the most significant development in neuroscience in the past decade, since it enables highly localized targeting at the single-cell level both *in vitro* and *in vivo*²⁰. Its reliance on genetic transfection introduces challenges and limitations, however, which has motivated extensive exploration of nongenetic means of optical control. Several reports have shown the possibility to achieve light-induced manipulation of cells, particularly excitable cells, either mediated by light-absorbing particles^{19,21,22}, thin-films²³⁻²⁵, or using direct near-infrared optical heating²⁶. In the past years, a growing spectrum of novel bioelectronics applications have been enabled by organic semiconductors which have superior biocompatibility and mechanical properties, and novel functionality relative to silicon²⁷⁻²⁹. These features, combined with their high optical absorbance coefficient, have made nanoscale thin films of organic semiconductors suitable for optoelectronic photostimulation of single-cells³⁰⁻³² and retinal tissues^{25,33}. The issue of the cell/semiconductor cleft still remains an obstacle for these organic devices, however. The starting point of our work is the desire to create a new family of organic semiconductor structures that can by virtue of morphology form an intimate contact with the cell membrane. To this end, we have synthesized hierarchical nanocrystalline colloidal architectures.”

Page 19...“Rapid and localized heat transfer to cells has been targeted by various applications both *in vitro* and *in vivo*^{19,26,31}, and clinical applications of optical neural stimulation for neuralprosthetics are currently actively explored⁵⁶.”

Line 319 on – the use of these materials for *in vivo* electrophysiology would rely on their conductivity. How conductive are these materials? How would their use be envisioned – for sensing or stimulation?

The conductivity of QNC has been reported for thin films in a few papers, as well as some work on doping to modulate conductivity. Mobility is in the range of 0.1 cm²/Vs for both electrons and holes. We have elaborated in the introduction section:

Page 4...“Recently QNC, in the form of vacuum-evaporated thin films, has been reported as a promising semiconductor, with outstanding stability, ambipolar charge carrier mobility, and favourable optoelectronic⁴⁸ and photocatalytic⁴⁹ properties. The multifunctionality and availability of this material made it a good target for making organic semiconducting hierarchical nanostructures.”

Line 324 – a tattoo is not an implant

Thanks for this comment – we have now removed this.

REVIEWERS' COMMENTS:

Reviewer #1 (Remarks to the Author):

This referee has carefully checked the rebuttal letter and the revised manuscript. The questions we raised in the initial review have been properly addressed with a substantial amount of experimental evidence. This referee would suggest this manuscript to be accepted in the current form.

Reviewer #2 (Remarks to the Author):

Review

Cellular interfaces with biomimetic hydrogen-bonded organic semiconductor hierarchical nanocrystals
Sytnyk et al

I congratulate the authors on their extensive revisions and on their paper which is now significantly stronger and more impactful.

There are still a few minor issues I see related to terminology which could be corrected. However I recommend acceptance with minor revision.

"271 RBLs, HEKs form a high interfacial area contact with hedgehog crystals, with evidence of extensive invagination of the cell membrane around the sharp nanocrystallites (Figure 5),"

I think invagination is not the correct term to use. I greatly appreciated the images in Figure 6 which show to me that the cells are spreading on the hedgehogs as if on a surface. Invagination for me refers more to a part of the cell displacing its membrane around a smaller obstacle. Most of the images in figure 6 show a planar membrane spreading very close to a hedgehog. The absence of cleft is indeed remarkable. The only evidence of engulfment which almost looks like phagocytosis is in Figure 5b – however it is not clear if it is a single cell or if there may be another cell below

« The hierarchical semiconductor nanoarchitectures shown here are a new platform for inducing photoeffects in a highly local fashion, providing a nongenetic optical means of changing channel activity of single cells.»

I would qualify this statement somewhat, as there will always be a lack of specificity compared to optogenetics methods unless a transfected cell line is used (such as the TRPV1 HEK cells) which obviously needs genetic approaches.

Finally, I would be very careful of equating strength of cell adhesion with biocompatibility. Often cells need to be able to move dynamically and bind and unbind to achieve their function.

We have implemented in the next round the requested corrections. In the following, you can find a point-by-point response to referee comments:

Reviewer comments: RED

Our response: BLUE

Text added to revised manuscript: "BLACK"

Reviewer #1 (Remarks to the Author):

This referee has carefully checked the rebuttal letter and the revised manuscript. The questions we raised in the initial review have been properly addressed with a substantial amount of experimental evidence. This referee would suggest this manuscript to be accepted in the current form.

Thank you kindly!

Reviewer #2 (Remarks to the Author):

I congratulate the authors on their extensive revisions and on their paper which is now significantly stronger and more impactful.

Thank you for this appreciation of the work we have done.

There are still a few minor issues I see related to terminology which could be corrected. However I recommend acceptance with minor revision.

"271 RBLs, HEKs form a high interfacial area contact with hedgehog crystals, with evidence of extensive invagination of the cell membrane around the sharp nanocrystallites (Figure 5),"

I think invagination is not the correct term to use. I greatly appreciated the images in Figure 6 which show to me that the cells are spreading on the hedgehogs as if on a surface. Invagination for me refers more to a part of the cell displacing its membrane around a smaller obstacle. Most of the images in figure 6 show a planar membrane spreading very close to a hedgehog. The absence of cleft is indeed remarkable. The only evidence of engulfment which almost looks like phagocytosis is in Figure 5b – however it is not clear if it is a single cell or if there may be another cell below

Thanks for this critical consideration. Agreed, the term invagination is not appropriate here since as you point out the dominant process is the planar membrane spreading very close to the hedgehog and the latter mechanically giving way to the cell as it grows. We have removed the term entirely and have modified the text accordingly. As you point out, we do present cases where engulfment-like events are visible, and their appearance is similar to some of the cited literature on nanoscale electrophysiology electrode probes, *e.g.* ref. 15, therefore we discuss this as a potentially important consideration and parallel to what has been observed by other researchers.

"From SEM imaging, it is apparent that the plasma membrane and extracellular matrix conforms to the nanoneedle structures, causing anchoring of cells onto the microhedgehog (Figure 4)."

"Like RBLs, HEKs form a high interfacial area contact with hedgehog crystals, (Figure 5), with occasional examples of engulfment-type processes¹⁵ clearly occurring (Figure 5b)."

« The hierarchical semiconductor nanoarchitectures shown here are a new platform for inducing

photoeffects in a highly local fashion, providing a nongenetic optical means of changing channel activity of single cells.”

I would qualify this statement somewhat, as there will always be a lack of specificity compared to optogenetics methods unless a transfected cell line is used (such as the TRPV1 HEK cells) which obviously needs genetic approaches.

Agreed, this statement is misleading and should be qualified. We are presenting a means of changing the channel activity that is of non-genetic origin, *i.e.* one is not transfecting anything with light-sensitive channels as in the case of optogenetic methods. What we want to underscore is the local nature of stimulation, coming from the fact that one has a close interface between a cell and a particle with similar size and shape as the cell and the fact that optical stimulation is afforded by a mechanism which is not coupled with the requirement for genetic transfection. Indeed as you say that specificity in terms of which ion channels are being stimulated does require transfected cell lines, and therein was the confusing aspect of the sentence as we originally wrote it. We have thus modified this concluding sentence (last sentence in *Photostimulation* section):

“The hierarchical semiconductor nanoarchitectures shown here, by virtue of being of similar size and shape to single cells, function as a platform for inducing photoeffects in a highly local fashion, providing means of stimulating cells without invoking the need for genetic modification with light-sensitive ion channels.”

Finally, I would be very careful of equating strength of cell adhesion with biocompatibility. Often cells need to be able to move dynamically and bind and unbind to achieve their function.

Agreed, these two parameters should not be equated. We do not believe that we made this type of claim in the manuscript - We can only say that in the case of HEK and RBL cells in our experiments we do not find any changes in viability versus control samples, and cite the literature on the nontoxicity of quinacridone as a consumer material. However, we believe that this highlighted sentence may be what was misleading:

“Over longer times in culture, the cells are found to transition from a rounded shape to spread more extensively on the hedgehogs (Figure 4d). This progressive behaviour is consistent with the interpretation that the cells are viable while having such an extensive contact area with the crystals, which is further corroborated using the CytoTox-Glo luminescence-based assay (Supplementary figure 14). Carried out over three days, the assay demonstrated no difference in viability between cells cultured with hedgehogs, planar or powder QNC, and control samples. This experiment suggests that neither form of the QNC material is acutely cytotoxic. QNC itself, as a commercial pigment, has been studied with regards to consumer safety and determined to be nontoxic⁴⁶.”

Therefore in the text we have now modified this to read as follows:

“Over longer times in culture, the cells are found to transition from a rounded shape to spread more extensively on the hedgehogs (Figure 4d). Based on viability assays (CytoTox-Glo luminescence-based assay, Supplementary Figure 14) we conclude that the cells remain viable while having such an extensive contact area with the crystals. Carried out over three days, the assay demonstrated no difference in viability between cells cultured with hedgehogs, planar or powder QNC, and control samples...”